# A novel endocast technique providing a 3D quantitative analysis of the gastrovascular system in *Rhizostoma pulmo*: An unexpected through-gut in cnidaria

**Massimo Avian**[1]☉*, **Lucia Mancini**[2¤a], **Marco Voltolini**[2¤b], **Delphine Bonnet**[3], **Diego Dreossi**[2], **Vanessa Macaluso**[1], **Nicole Pillepich**[1], **Laura Prieto**[4], **Andreja Ramšak**[5], **Antonio Terlizzi**[1,6], **Gregorio Motta**[1,6]☉

**1** Department of Life Science, University of Trieste, Trieste, Italy, **2** Elettra-Sincrotrone Trieste S.C.p.A., Basovizza, Trieste, Italy, **3** MARBEC, Université de Montpellier, CNRS, Ifremer, IRD, Montpellier, France, **4** Group Ecosystem Oceanography, Department of Ecology and Coastal Management, Instituto de Ciencias Marinas de Andalucia (CSIC), Cádiz, Spain, **5** National Institute of Biology, Marine Biology Station, Piran, Slovenia, **6** Department of Integrative Marine Ecology (EMI), Stazione Zoologica Anton Dohrn, Napoli, Italy

☉ These authors contributed equally to this work.
¤a Current address: Slovenian National Building and Civil Engineering Institute, Ljubljana, Slovenia
¤b Current address: Department of Earth Science Ardito Desio, University of Milano, Milano, Italy
* avian@units.it

**Data Availability Statement:** All relevant data are within the paper and its Supporting information files.

## Abstract

The investigation of jellyfish gastrovascular systems mainly focused on stain injections and dissections, negatively affected by thickness and opacity of the mesoglea. Therefore, descriptions are incomplete and data about tridimensional structures are scarce. In this work, morphological and functional anatomy of the gastrovascular system of *Rhizostoma pulmo* (Macri 1778) was investigated in detail with innovative techniques: resin endocasts and 3D X-ray computed microtomography. The gastrovascular system consists of a series of branching canals ending with numerous openings within the frilled margins of the oral arms. Canals presented a peculiar double hemi-canal structure with a medial adhesion area which separates centrifugal and centripetal flows. The inward flow involves only the "mouth" openings on the internal wing of the oral arm and relative hemi-canals, while the outward flow involves only the two outermost wings' hemi-canals and relative "anal" openings on the external oral arm. The openings differentiation recalls the functional characteristics of a through-gut apparatus. We cannot define the gastrovascular system in *Rhizostoma pulmo* as a traditional through-gut, rather an example of adaptive convergence, that partially invalidates the paradigm of a single oral opening with both the uptake and excrete function.

## Introduction

Cnidaria is a Precambrian phylum comprising about 10.000 living known species, and nearly 4,055 of these are Medusozoa [1]. Nowadays, jellyfish have become increasingly important for the scientific community as jellyfish blooms have been observed to increase all over the world

**Funding:** The authors received no specific funding for this work.

**Competing interests:** The authors have declared that no competing interests exist.

[2–4]. Jellyfish can play a dominant role in structuring planktonic communities, and they may directly and indirectly interact and interfere with human economic and recreational activities, ecosystem services, public health and local wildlife [5]. Thus, a deep knowledge about their biology, physiology, anatomy and ecology is required to understand the dynamics of blooms and to predict their occurrence and impacts, also in relation to climate change.

Within Cnidaria, the gastrovascular apparatus is based mainly of a central cavity, the stomach, connected with the outside through a single opening, the "mouth", and from which a series of extensions, pouches or canals, branch out, reaching the periphery of the organism. The internal fluid circulation is ensured by the ciliary motion of the gastrodermic layer that delimits the "gastrovascular" system. To provide a constant exchange of internal fluids, a double circulation (in and out) has therefore developed, both in the anatomical polyp body plan (with the formation of one or two ciliated furrows, the siphonoglyphs, in several anthozoans), and in the medusa one [6–8].

The Medusozoa are a monophyletic group with a wide range of gut anatomies and digestive mechanisms [9]. This Subphylum is characterized by a sac-like gut with a single opening that acts both as mouth and anus [6, 7]. Ciliary currents carry food to the gut through the mouth as well as perform food egestion from the same oral opening [6]. This general idea is supported by anatomical and physiological studies [10–12]. Within the clade Acraspeda, staurozoans maintain a roughly polyp-like circulation pattern even in the benthic medusa stage, whilst in the Rhopaliophora (Cubozoa and Scyphozoa) clade jellyfish exhibit a series of pouches (Cubozoa; Coronatae, Pelagiidae, Cyaneidae, Drymonematidae and Phacellophoridae within Semaeostomeae scyphozoans) or canals (Ulmaridae within Semaeostomeae, and Rhizostomeae) reaching the umbrella margins. However, most of the jellyfish maintain a single central opening, the cruciform mouth. In gastrovascular systems thus structured, the inward and outward circulation begins to be spatially separated with the appearance of the umbellar canals in the Ulmaridae, characterized by centrifugal flows in the adradial canals, and centripetal flows in the perradial canals. In the stomach, the outward perradial currents flow outside the umbrellar gastrovascular system from the peripheral base of the gastro-oral groove (= the edges of the cruciform manubrium canal) while the inward currents flow in the proximal part of the groove (= the medial portion of the cruciform manubrium canal [7, 13, 14]. A further increase in complexity, with the regression of the central mouth replaced by a network of oral arm canals (initially four, originating by the cruciform grooves of the original mouth, with a complicated course due to the hypertrophic development of the four genital sinuses) is present in the Rhizostomeae only. Some benthic Cambrian medusozoan fossils, very well preserved, probably cubozoans, are devoid of radial canals [15, 16], so the potential evolutionary steps could have been the extension of broad gastric pouches up to the umbrella margins (in swimming medusae), the reduction of broad gastric pouches by fusion between exumbrellar and subumbrellar gastroderm [15], their ramification, especially in the inter- and adradial areas (involved to centrifugal fluxes), and the reduction of the single central mouth with the origin of a network of oral arms canals, which led to the appearance of numerous mouths, a specialization for a planktophagous diet, in rhizostomean medusae. Within these adaptations, the network of oral arms canals and related "mouths" must therefore make up for both incoming and outgoing flows, and, to date, the most widespread idea (in the absence of experimental evidence) hypothesized that the direction of the ciliary currents within the oral arm canals may be reversible. According to this theory, the internal circulation rely on the same single opening (or many) for both ingestion and egestion of food. However, Arai and Chan (1989) demonstrated that the hydromedusan *Aequorea victoria* possess other openings in addition to the central mouth. *A. victoria* was shown to egest material from both its gastrovascular cavity through its mouth, but also through radial subumbrellar papillae and pores [17]. Despite this

is not being a proper trough-gut, it was the first observation that collides with the single oral opening paradigm of Cnidarian anatomy. Similarly, in Ctenophora, Presnell (2016) [18] proved that *Mnemiopsis leidyi* possess a proper unidirectional tripartite through-gut. This evidence deeply collides with the common theories of the evolution of the metazoan through-gut, although its origin is currently a matter of debate [19]. Furthermore, other recent studies are now discussing the hypothesis that Ctenophora may be the first branching extant metazoan phylum [20–23].

The common statements on the metazoan evolution take it for granted that the first through-gut appeared within Bilateria [24, 25]. Except for Porifera and Placozoa, the other non-Bilateria phylum such as Cnidaria and Ctenophora are thought to possess simple digestive systems for the extracellular breakdown of ingested food [26]. A hypothesis concerning the through-gut origin speculates that it may have appeared in the elder metazoans, and it may have been lost in successive phyla until Bilateria or ctenophores converged to an analogue organ close to the bilaterian through-gut [18]. However, at least in the cnidarians, and especially in the jellyfish body plan, it seems that the gastrovascular system has undergone various adaptations, some of which seem an adaptive convergence with the bilaterian through-gut, like the *Aequorea* group, and other taxa, which developed an innovative complex network of anastomosed canals both in the umbrella (similar to a capillary network) and the oral arms, with relative increase of the openings ("mouths") with the outside.

*Rhizostoma pulmo* (S1 Fig) is one of the first jellyfish species formally described in history [27], however, up to now, a complete description of its gastrovascular system is still missing, despite being a rather common jellyfish along the coasts of the Mediterranean Sea [28, 29]. As part of a research on defining the phylogenetic pattern of the *Rhizostoma* genus, the morphology and functional anatomy of the species *R. pulmo* were analyzed in detail.

In this work, we investigated the morphology of the gastrovascular system of jellyfish by using a novel approach combining resin endocasts and laboratory-based X-ray computed microtomography (μCT). The μCT data acquisition of the cast replicating the gastrovascular system of the medusa opened, for the first time, the opportunity to describe its manubrium system topology in a fully quantitative fashion. The comprehension of the gastrovascular system and its circulation pattern are very important to understand jellyfish feeding physiology and their role in the trophic webs.

Our main objectives were:

1. to create a new protocol for the investigation of the morphology of gastrovascular systems in jellyfish by using epoxy resins in order to create solid three-dimensional (3D) casts of the vascular systems.

2. to describe and measure the morphology of the umbrellar and manubrium gastrovascular system from X-ray μCT.

3. to investigate the circulation pattern among the complex canal system of *R. pulmo*.

## Results

### Morphological analyses

As a result of contrast injections (see S3 Fig), cast experiments (Fig 1a) and X-ray μCT (Fig 1b), it has been possible to analyze the gastrovascular structure of *Rhizostoma pulmo*. The present study allowed us to provide new and original morphological-anatomical details both at umbrellar and manubrium level. All diameter values in this work refer to the jellyfish apparent diameter, not the real diameter (see Methods, S1a Fig).

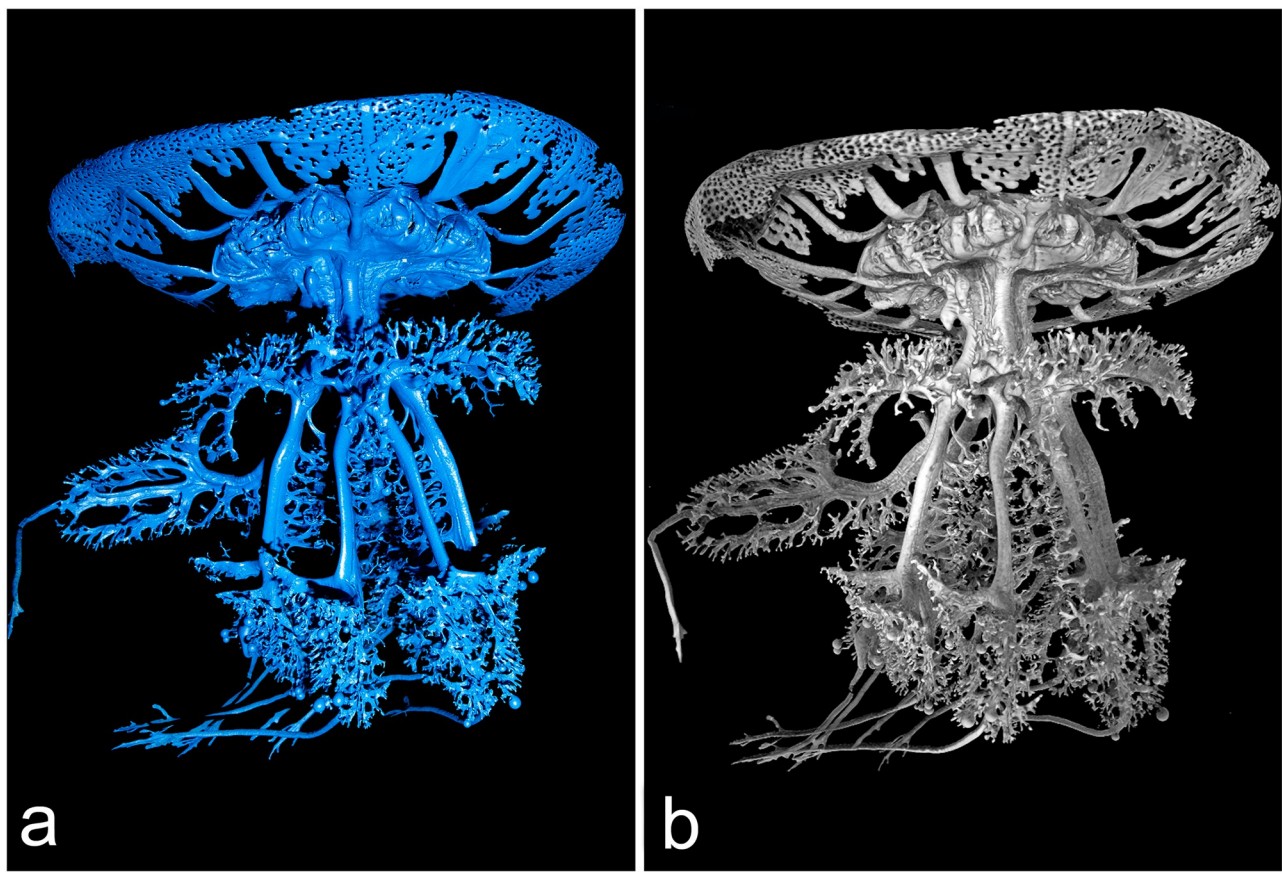

**Fig 1. *Rhizostoma pulmo* gastrovascular system of the manubrium.** (a) Resin endocast of the whole gastrovascular system of a specimen of 14.6 cm in diameter. (b) 3D rendering obtained by X-ray microtomographic data of the resin endocast showed in (a). Tomographic reconstruction performed with an isotropic voxel size of 62.0 μm.

**Umbrella.** At umbrellar level, the stomach is four sided, from which sixteen radial canals originate. Four cross shaped straight perradial canals originating from the peripheral ring canal are literally immersed in the subumbrellar mesoglea layer (Fig 2).

In specimens of about 5–10 cm in diameter, the two subumbrellar edges that delimit these canals from the stomach are in slight contact, starting to virtually isolate them (Fig 3a and 3b). In larger medusae these edges adhere more widely, making the perradial canals practically independent from the overlying stomach (Fig 3c and 3d). They are alternated with the four interradial canals as well the eight adradial canals. All inter- adradial canals origin from the upper part of the stomach, then they bend downwards before getting straight in the distal umbrellar portion, just over the coronal muscle (Figs 2, 3c, 4b).

The endocasts also showed that the stomach roof is not concave, but convex (Fig 4a–4d).

This central upper convexity can be better seen in younger specimens, less than 15 cm in diameter. Later on, this convexity develops a central, protruding, flattened four-sided pyramid that continues distally with four small ridges exactly corresponding to the perradia (Fig 4c and 4d).

The stomach floor is cruciform, composed by four interradial triangular areas, alternated with four perradial depressions which correspond to the perradial canals (Fig 3a and 3b, S2c and S2d Fig).

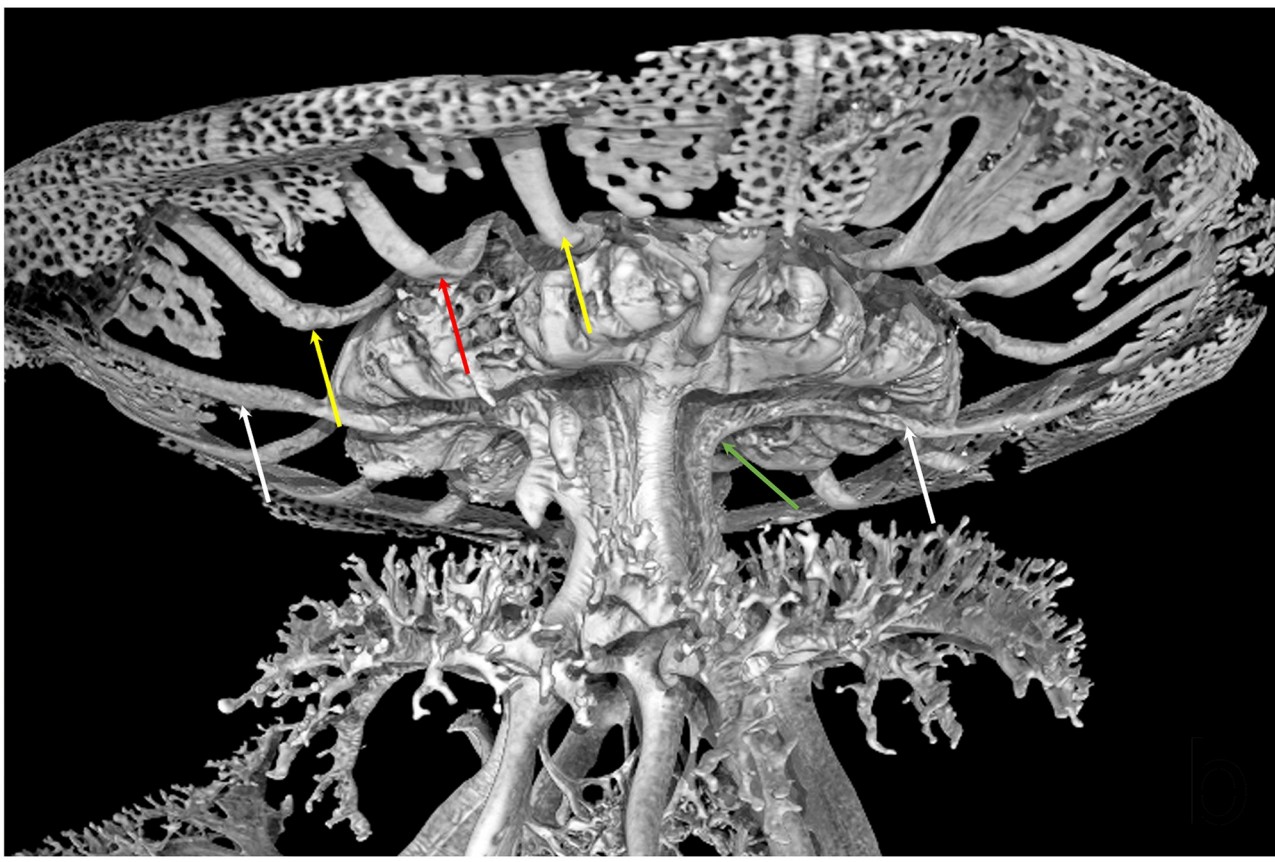

**Fig 2. Volume rendering of the resin endocast obtained via X-ray µCT, showing an enlarged view of Fig 1b: Detail of the stomach, subumbrellar view.** The radial canals per quadrant, emerging from the stomach are visible. Arrows indicate: two perradial canals (white), one interradial canal (red), two adradial canals (yellow) and the perradial canal passage into the manubrium (green). CT reconstruction performed with an isotropic voxel size of 62.0 µm.

**Manubrium.** The gastrovascular system of the manubrium starts with a deep cruciform opening, the edges of which correspond to the perradial canals. This cruciform opening is the residual of the central mouth of the ephyra—post-ephyra stages (Figs 3a and 3b, 5a and 5b, 6a–6d, S2c and S2d Fig). In adults, the mouth opening, still open in the post-ephyra phase (up to a diam. of about 3 cm, S2a and S2b Fig), is almost sealed (S3a Fig).

From the edges of this cruciform opening originate four apparently flattened canals. The canals section is not simply flattened, rather recall an eight shape, compressed in the median portion. This portion is not fused (as the stains injections in fixed samples seemed to evidence, S3b Fig), but the gastrodermic layer of both sides is in contact through a specular crenulation (Fig 5c and 5d). This structure simulates a double canal system. In living specimens, these adhesions limit the communication between the two hemi-canals. Comparing the same canal's cross-sections of a specimen with a diameter of 18 cm with an adult specimen of 26.1 cm, and calculating an "adherence ratio" (adhering stripe width/canal total width) (Fig 5c and 5d) it was possible to observe that the ratio remains rather constant, with a value of 0.35 for the young specimen, and of 0.36 for the adult one, even if the adhering stripe width increases from 2.88 mm to 3.53 mm in the adult.

In the proximal portion of the manubrium, between the genital sinuses and the scapulae (= scapulets, epaulettes), the outer hemi-canal bifurcates into two, while the inner hemi-canal

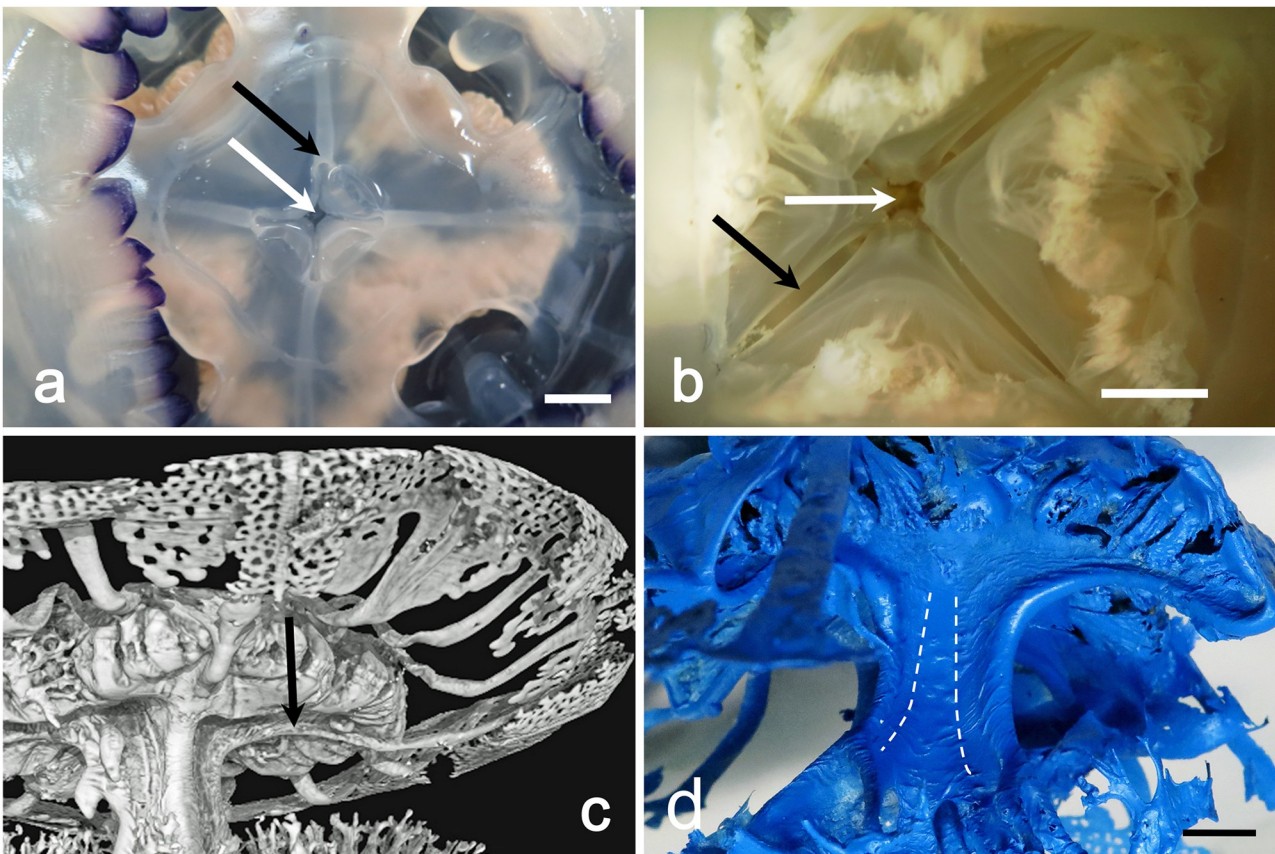

**Fig 3. Relationship of the perradial canals and the juvenile mouth opening.** (a) Subumbrellar view of a young medusa, manubrium excised. White arrow indicates the remnant of the original mouth. Black arrow indicates the section of one perradial canal. The edges that delimit the canals are still partially adjacent (scale bar = 1 cm). (b) Exumbrellar view of the stomach floor of another young specimen. White arrow indicates the quadrangular central remnant of the mouth. Black arrow indicates the still partially opened edges of a perradial canal (scale bar = 0.5 cm). (c) Volume rendering of the subumbrellar region of the resin endocast, obtained by X-ray µCT. Black arrow indicates a protruding perradial canal from the stomach floor. CT reconstruction performed with an isotropic voxel size of 62.0 µm. (d) Endocast of a larger specimen (35 cm diam.). The area of adherence of the perradial canal edges is noticeably increased. Dashed lines indicate the end of the adhering stripes in the proximal portion of the manubrium (scale bar = 1 cm).

bifurcates just below (two overlapped "Y"-shape branches, Fig 5a and 5b), forming a total of eight hemi-canals, one for each oral arm. At scapulae level (each oral arm has two scapulae), two hemi-canals (the distal and the proximal) bifurcates into two lateral hemi-canals which continue into the scapulae (Fig 5e–5g). The endocast also highlighted the pattern of the scapular canal emergence from the oral arm canals. The disposition of the upper secondary ramified scapular canalicula seems mostly related with the upper hemi-canal. The most distal, always dichotomously branched, is connected with the lower hemi-canal. This canalicular pattern undergoes further complications in larger specimens, where some lateral canaliculi also connect with the lower hemi-canal (Fig 5f and 5g).

Within each scapula the canals branch into many tiny canals reaching the upper frilled margins, ending in multiple openings (Fig 5g).

However, it was noticed that, especially in the smaller ones, like that of 14.6 cm in diameter measured by X-ray µCT measurements (Figs 1, 3c, 4a, 5e), the resin injected into the gastrovascular system tends to slightly dilate the shape of the canals, reducing or masking the medial

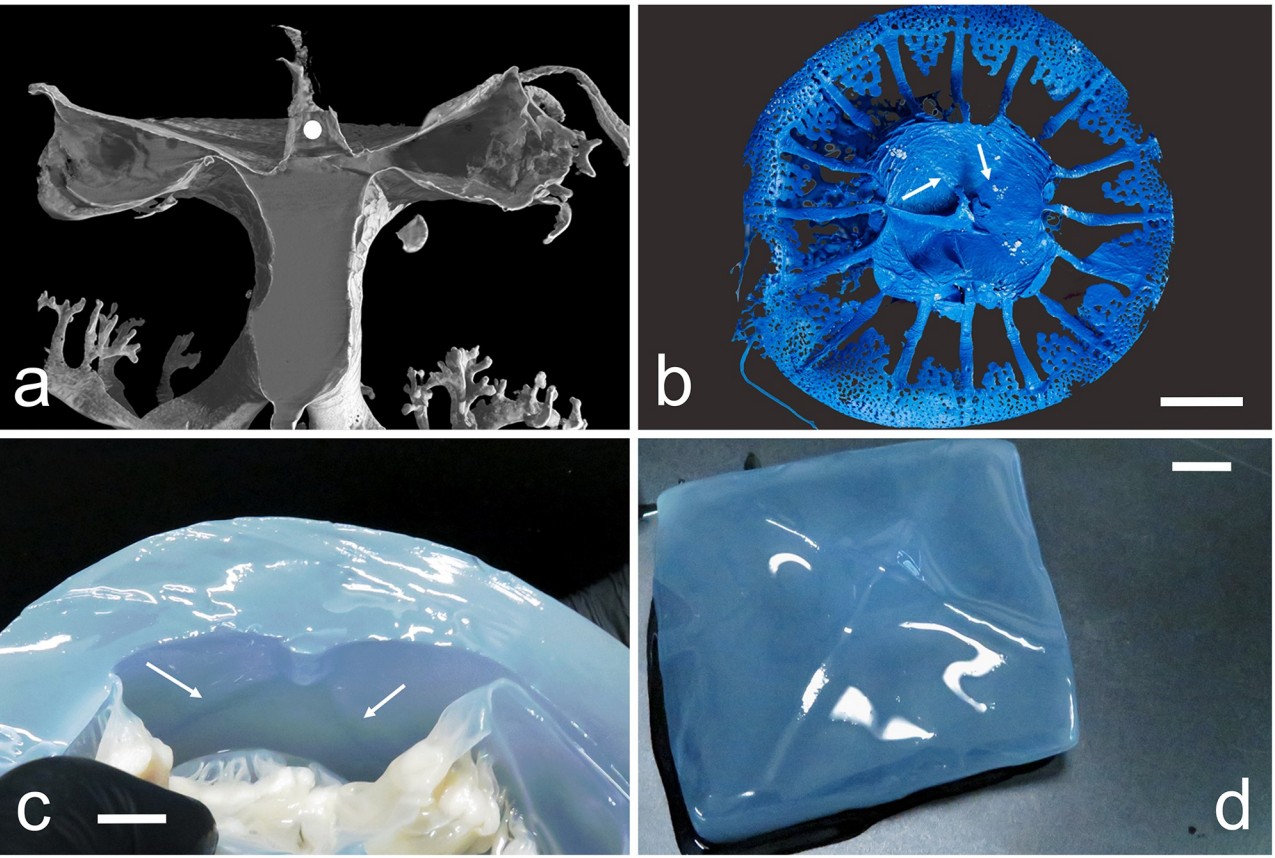

**Fig 4. Morphology of the exumbrellar limit of the stomach.** (a) Median sagittal section of the endocast showed in Fig 1b, evidencing the convexity of the stomach roof. White spot indicates the artefact due to the resin injection. CT reconstruction performed with an isotropic voxel size of 62.0 μm. (b) Endocast exumbrellar view. White arrows indicate two of the four perradial outlined ridges (scale bar = 1 cm). (c) Interradial section of a 29 cm adult evidencing the upper central protruding pyramid. Arrows indicate two of the perradial ridges (scale bar = 1.5 cm). (d) Subumbrellar view of the stomach roof of a specimen of 33 cm diam. showing the central protruding square pyramid and the four perradial ridges (scale bar = 1 cm).

narrowing present in most of the canal system. In larger specimens (diam. $\geq$20 cm) this effect is strongly reduced, and, in some cases, it has been observed that the resin forms a very thin layer in the median area of adherence, thus making the two distal portions of the canals appearing clearly thicker (Figs 3d, 5f–5h). In larger medusae the origin of the lower scapular hemi-canal starts from the outer, distal oral arm hemi-canal, while the upper one starts from the internal, proximal hemi-canal of the oral arm. The adhering, medial portion of the scapular canal, begins from the corresponding adhering area of the oral arm canal, slightly sloped (Fig 5f). This canal pattern is symmetrical, since each oral arm gives rise to two scapulae, one on each side (Fig 5g).

The other two hemi-canals continue under the scapulae (Fig 5b and 5h) to the three-winged portion of the oral arm (S1b Fig). Here, in the initial part, in specimens with a diameter smaller than 15–20 cm, the two hemi-canals anastomose and subsequently fuse in the central part (Fig 5i, S3b Fig) sending many lateral branches to the marginal opening of each wing. The fused canal continues into the terminal club, where it branches again at its ending (Fig 5i). In larger medusae (diam. > 20 cm), however, the anastomosed tract regresses, and the two hemi-canals remain separated until the beginning of the terminal club canal, which is the only segment that remains undivided.

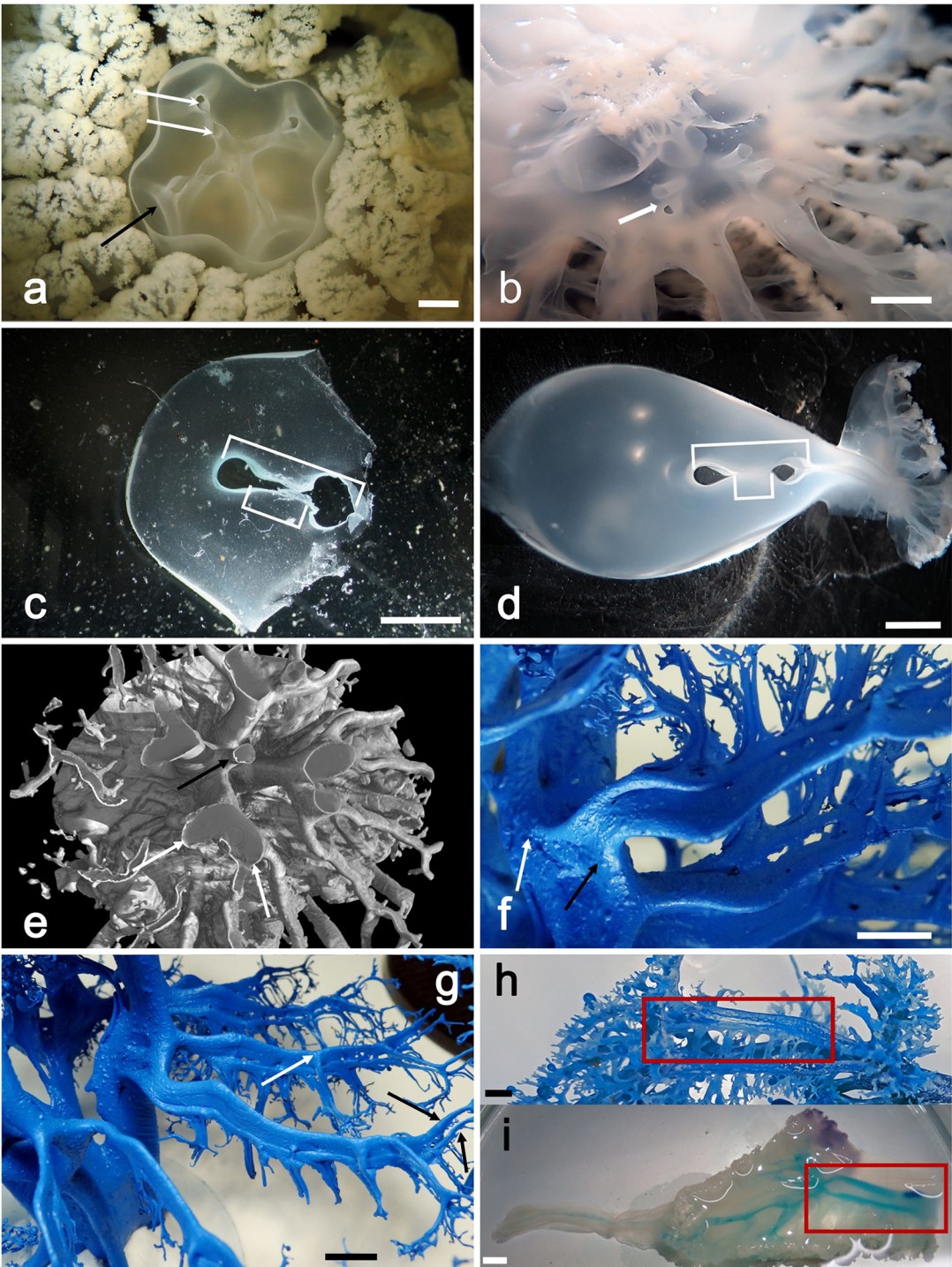

**Fig 5. Morphology of the manubrium gastrovascular system.** (a) Manubrium section over the emergence of the scapulae, exumbrellar view. Visible the hemi-canals pattern (white arrows) branching from the central cavity, the internal one partially emerging. Black arrow indicates one of the further dichotomic branches of the oral arms duplication (scale bar = 0.5 cm). (b) Section at subscapular level, subumbrellar view (scale bar = 1 cm). Arrow indicates one of the hemi-canal structures. (c) Transverse section of an oral arm at sub scapular level. Larger square bracket indicates the total canal length, smaller one the medial adhering area (scale bar = 0.5 cm). (d) Same as in (c) (scale bar = 0.5 cm). (e) Subumbrellar 3D rendering of a specimen sub-volume obtained by X-ray μCT; white arrows indicate the dichotomization of two oral arms main canals, black arrow indicates the remnant of the juvenile central mouth opening. CT reconstruction performed with an isotropic voxel size of 62.0 μm. (f) Endocast of a specimen of 32 cm diam. showing the emergence of the scapular hemi-canal. Arrows indicate the

upper (white) and lower (black) hemi-canal (scale bar = 1 cm). (g) Scapular canal system, upside down view. White arrow indicates one branching connected to the lower canal, black arrows indicate the terminal dichotomic branching maintaining the two hemi-canals pattern (scale bar = 1 cm). (h) Endocast of one oral arm, showing the two hemi-canals pattern (scale bar = 1 cm). (i) Stain injected in the oral arm evidences the hemi-canal pattern. Red squares indicate the correspondent trait as in (h). The juvenile anastomoses on the left will disappear in more developed medusae (scale bar = 1 cm).

**Image processing and analysis of the X-ray μCT data.** The most complete endocast from a specimen of 14.6 cm in diameter was firstly visualized by means of virtual sectioning and using 3D rendering procedures (Fig 1b).

In addition to the computed morphometric analysis of the tomographic data (discussed in the next paragraph), the X-ray μCT data also allowed the investigation of the less accessible areas of the gastrovascular system, impossible to be properly examined by visual observations. The endocasts analyses already evidenced that the remnant of the juvenile central mouth opening (the central canal) continues with a small canal that apically branches into a series of small distal canals and relative openings right in the center of the oral arms (Figs 5e, 6a–6d, S1–S4 Videos). However, its position and the complexity of the surrounding canals pattern masked this area, thus making it impossible to have a clear picture of this structure, and to describe it in detail.

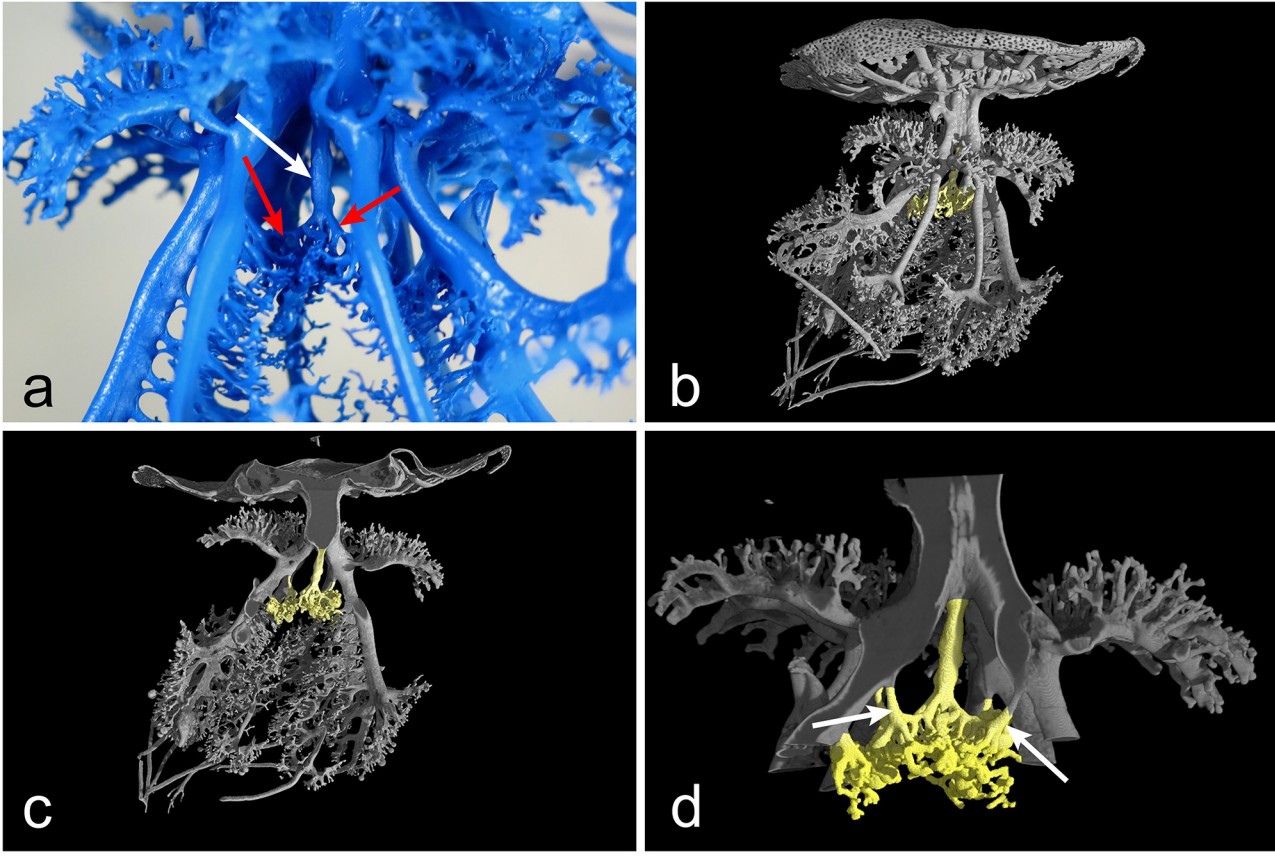

**Fig 6. Morphology of the manubrium gastrovascular system.** (a) Proximal internal region of the manubrium, showing the projection of the juvenile central mouth (white arrow), distally branching into a series of small canals with terminal openings (red arrows). (b) Volume rendering obtained by X-ray μCT showing the hidden central canal and its branches (yellow). (c) Longitudinal section of the volume shown in (b), highlighting two lateral canals connected with two oral arms. (d) Magnification of the region showed in (c), featuring all the four transverse connections with the proximal bifurcations of the emerging eight oral arms (arrows indicate two of them). CT reconstruction performed with an isotropic voxel size of 62.0 μm.

The 3D rendering and the consequent longitudinal sections highlighted that, in correspondence of the dichotomy of the oral arms (from the four proximal to eight distal) there are four transverse canals connecting the central canal with the point of emergence of the eight oral arms canals (Fig 6b–6d; S1–S4 Videos), slightly asymmetrically shifted to the left one. This connections pattern was observed in other endocasts of larger specimens, but it was slightly different, both in number (maybe two or three) and symmetry (some connections seem to finish exactly in the middle of the dichotomic oral arm canals).

**Morphometric analysis of the tomographic data.** A graphical summary of the morphometric analysis of the tomographic data is shown in Fig 7.

The total volume of the gastrovascular system and the single volumes of umbrella, scapulae, central canal and oral arms of the 3D digitalized endocast (specimen of 14.6 cm in diameter), are of 26.050 cm$^3$, 11.591 cm$^3$, 5.955 cm$^3$, 0.193 cm$^3$ and 8.312 cm$^3$, respectively.

The volume rendering of the whole cast measured via X-ray µCT shown in (a), the yellow circle highlights the arm used for the more specific analysis of the single oral arm reported in

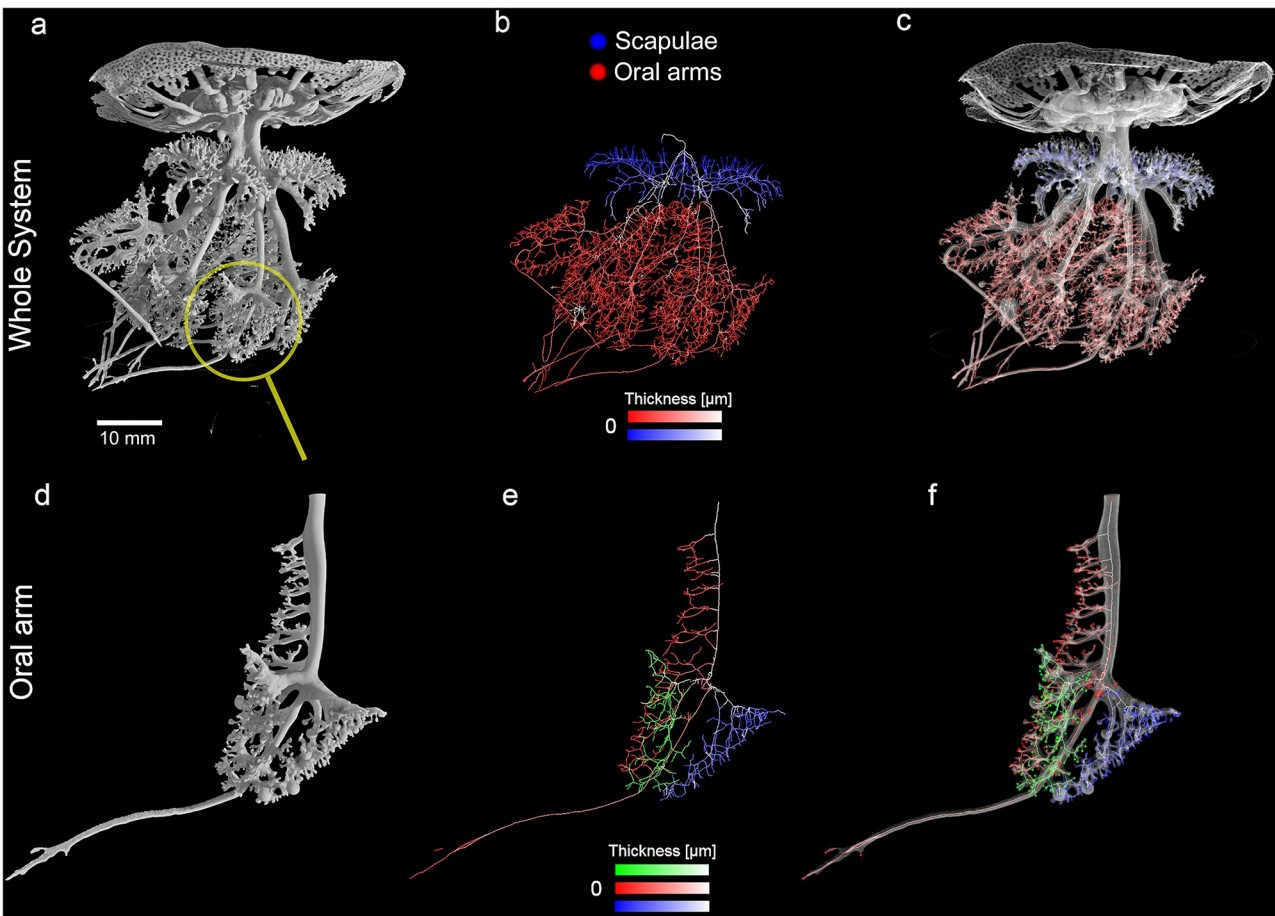

**Fig 7. Graphical summary of the results of the quantitative analysis on the gastrovascular system.** (a) 3D rendering of the whole cast of the gastrovascular system measured via X-ray µCT. (b) Isosurface rendering showing the segmented gastrovascular system of the whole manubrium; false colors refer to the two different structures analyzed (blue = scapulae, red = oral arms). Color intensity is proportional to the local thickness values in the medial axis at that specific voxel. (c) Volume rendering of the cast; medial axes analysis, high intensity colors indicate the smaller canals. (d) Volume rendering of a single oral arm. (e) Thickness-labeled skeleton of the three-winged portions, each characterized by a specific hue (red, green, and blue). (f) Openings of the three different wings labeled with the three different colors as in (e). CT reconstruction performed with an isotropic voxel size of 62.0 µm.

(d). In (b), the hue refers to the two different structures analyzed (blue = scapulae, red = oral arms), the intensity being proportional to the local thickness values in the medial axis at that specific voxel (e.g., in the oral arms light pink indicates a wide canal, while the strong red shows the thinner canals). In (c), a highly transparent volume rendering of the cast and the canals pattern has been superimposed to better highlight the context of the medial axes analysis, with the high intensity colors being more distributed close to the distal openings, while the low intensity colors being localized in the large canals of the system, as expected. In the panels (d-f), the analysis of a single oral arm is shown, starting from the volume rendering (d), the thickness-labeled skeleton of the three-winged portions, each characterized by a specific hue (red, green, and blue) is observed, and again with the color intensity proportional to the thickness of the structure at each point (e). Finally, in (f) the skeleton has been superimposed to a transparent volume rendering of the arm, with the addition of the oral openings viewed as small spheres of pure red, green, and blue, depending on the part that the specific oral opening belongs to.

A complete analysis of the cast has been also carried out, addressing the differences of the oral arms vs. the scapulae systems. In Fig 8a, the frequency histogram of the branch lengths of

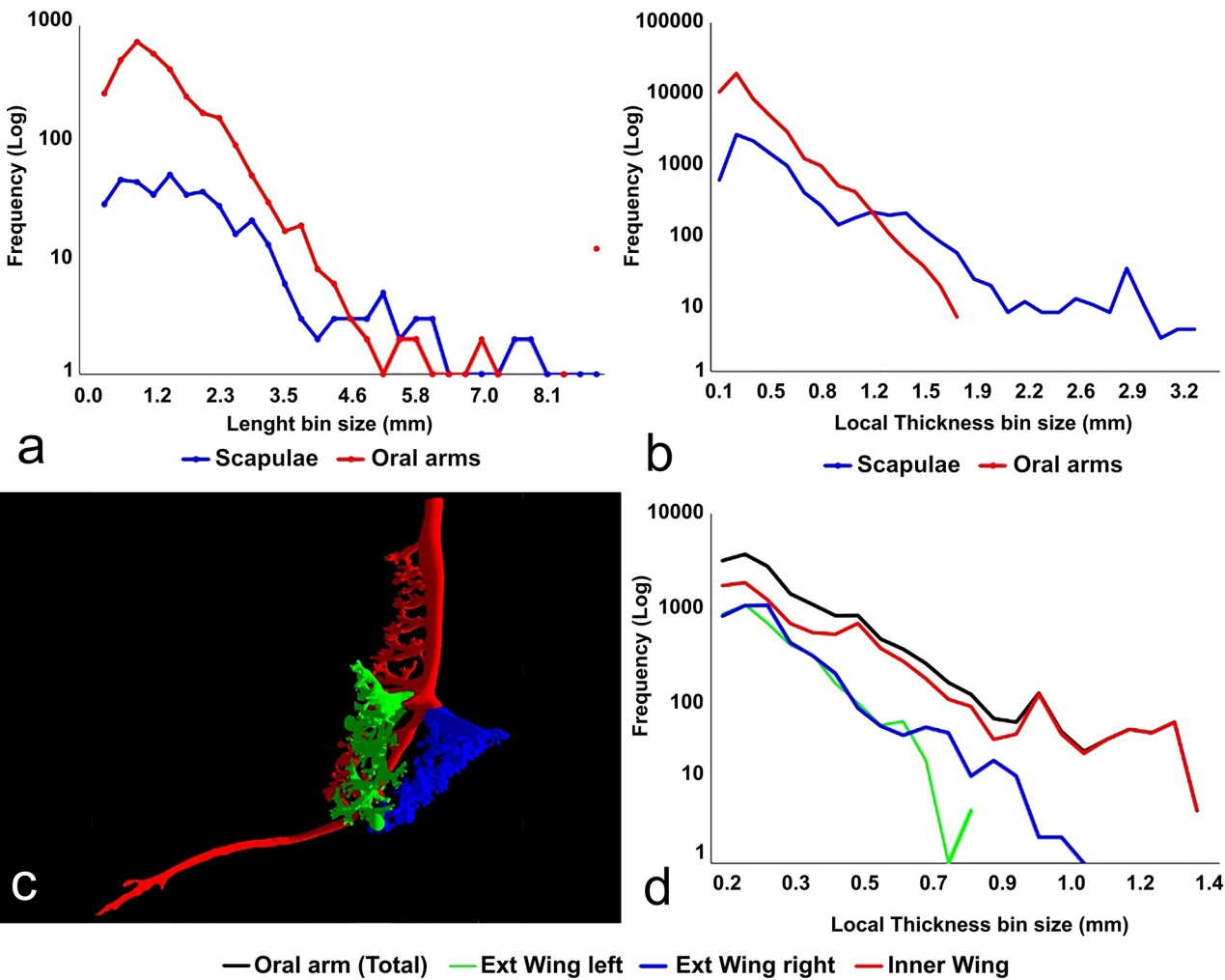

**Fig 8. Morphometric analysis of the scapulae vs. the oral arms and thickness analysis of the three-winged portions in an oral arm.** (a) Frequency plots of the branch lengths. (b) Thickness of the scapulae vs. the oral arms. (c) 3D isosurface rendering of a single oral arm. (d) Frequency histogram of the LT values within the three wings.

the two systems is plotted. As expected, the number of the branches is much higher in the oral arms, being a larger structure, but other interesting features, not easily recognizable from the graphical outputs, can be found: the mode of the branch length in the two systems is different, being higher in the scapulae than in the oral arms.

The distribution of the lengths in the scapulae is also significantly wider, with a less sharp peak and a less steep slope where the branch lengths increase (Fig 8b). The oral arms also show a small amount of very long branches, which are the canals in the terminal clubs.

Table 1 provides important morphological measurements about the length of the branches in the two different structures. Despite the long canals, the average branch length is noticeably smaller in the oral arms, with the more terminal portion of the canals being numerically the most important contributor of the gastrovascular structure (Fig 8a).

The tortuosity of the branches is almost identical (and rather straight), as highlighted by the Euclidean Distance of the two endpoints vs. the length of the branch being slightly higher than 0.8. More, the number of endpoints (roughly equivalent to the number of openings) vs. the number of branches is higher in the oral arm structure, highlighting again the larger relative contribution of these canals to the system.

Concerning the thickness analysis of the two systems, the frequency histograms of the local thickness (LT) values of the cast have been plotted in Fig 8b. The oral arms frequency distribution highlights an almost perfect lognormal distribution of the thicknesses. In the scapulae, the section of the plot featuring the smaller size of the canals shows almost exactly the same lognormal trend. The contribution (at the higher LT values) of the central main canals connecting to the stomach, splitting first in 4 and later in 8 channels, is evidenced by the peek in the frequency plot.

The analysis on the single oral arm aims at highlighting the differences of the two outer winged portions compared to the inner one. The investigated oral arm has been separated in three sub-structures, independently analyzed, as shown in Fig 8c.

In the thickness analysis of the single arm (Fig 8d), the two outer wings (green and blue) show trends almost exactly overlapping, while the inner wing (red) presents a slightly different trend due mostly to the presence of the terminal club canal and the thickest canals where the smaller ones are connected to it (see also Fig 7e).

In addition to thickness analysis, on the same arm (the best accurate reproduction of an oral arm), the number of the openings has been calculated. The number of endpoints in the inner (red) and external (blue and green) parts of the three-winged portion accounted for 234 (internal), 129 and 128 openings (external), respectively.

**Table 1. Summary of the skeleton analysis [43].**

|  | Scapulae | Oral Arms |
|---|---|---|
| Average Branch Length [mm] | 3.54 | 2.36 |
| Average Branch Euclidean Distance [mm] | 2.87 | 1.93 |
| ED/Length | 0.812 | 0.820 |
| Number of Total Branches | 396 | 3304 |
| Number of Total Junctions | 192 | 1685 |
| Number of Total Endpoints | 205 | 2987 |

Results from the computed morphometric analysis of the tomographic data from the endocast of a specimen of 14.6 cm in diameter.

ED = Euclidean Distance.

Likewise, the number of the openings of the central canal (Fig 6) has been quantified, with its branching ending with 81 endpoints.

Considering the whole organism (8 oral arms), based on the numbers reported above for the single oral arm, the inner openings (central canals + internal wing of the oral arms) should account for about 1953 openings while the external ones (blue and green parts of the winged portion) should account for about 2048.

**Functional anatomy.** The injections experiments in living specimens show a clear and permanent two-way flow circulation within the two hemi-canals. After injecting the staining solutions in the center of the stomach, the stain initially flows only in two directions, down towards the manubrium and into the umbrellar canals.

In the umbrella, the stain flows outwards only in the adradial canals first (Fig 9a and 9c), reaching the proximal main ring canal. From this ring, the stain circulates via the inner ring canal either towards the interradial and perradial canals (Fig 9a–9c), and towards the distal ring canal (Fig 9d), starting to stain the inner anastomosed canals mesh (Fig 9e). When the distal ring canal is reached, the stain flows towards the outer portion of the interradial and perradial canals, staining the rest of the canals mesh (Fig 9f; S5 Video). The stain into the interradial canals flows into the upper portion of the stomach again, while the stain into the perradial canals reaches the edges of the central stomach cruciform opening, driven towards the manubrium (Fig 9a–9f; Fig 11a).

At manubrium level, the stain solution evidenced a descending stream in the distal, outer hemi-canals (Figs 9b and 10a) and an initial spill of a mucous colored filament is evicted mainly from the openings placed in the upper portion of the external oral arm wings (Fig 10a).

A second experiment was performed on a living but excised oral arm at subscapular level of an adult specimen (diameter of 27 cm) where the stain was placed right on the cutting surface, showed an incoming flow only in the external, distal hemi-canal. This flow travelled up to the

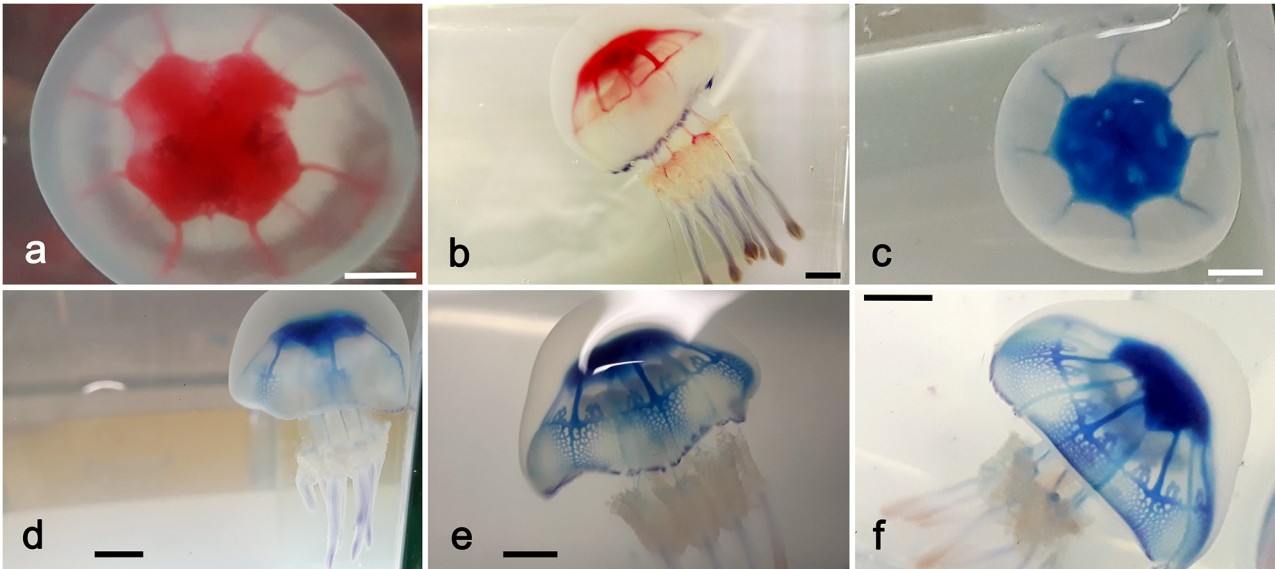

**Fig 9. Functional anatomy of the gastrovascular system.** (a) Living specimen with stained stomach, with stain diffusion into the adradial canals after 1 min from the injection (scale bar = 1 cm). (b) Same specimen, lateral view, after 1 min from (a). Staining of the internal ring canal. Manubrium: the stain transits into the external hemi-canals, and the upper openings of the external oral arm wings (scale bar = 1 cm). (c) Similar to (a), but specimen injected with methylene blue stain (scale bar = 1 cm). Adradial canals stained. (d) Initial coloration of the inner ring canal and the outwards adjacent anastomosed canals mesh (scale bar = 1 cm). (e) Beginning of coloration of the per- and interradial canals inside the inner ring canal. Beginning of coloration of the distal ring canal (scale bar = 1 cm). (f) Umbrellar canal system now completely stained (scale bar = 1 cm).

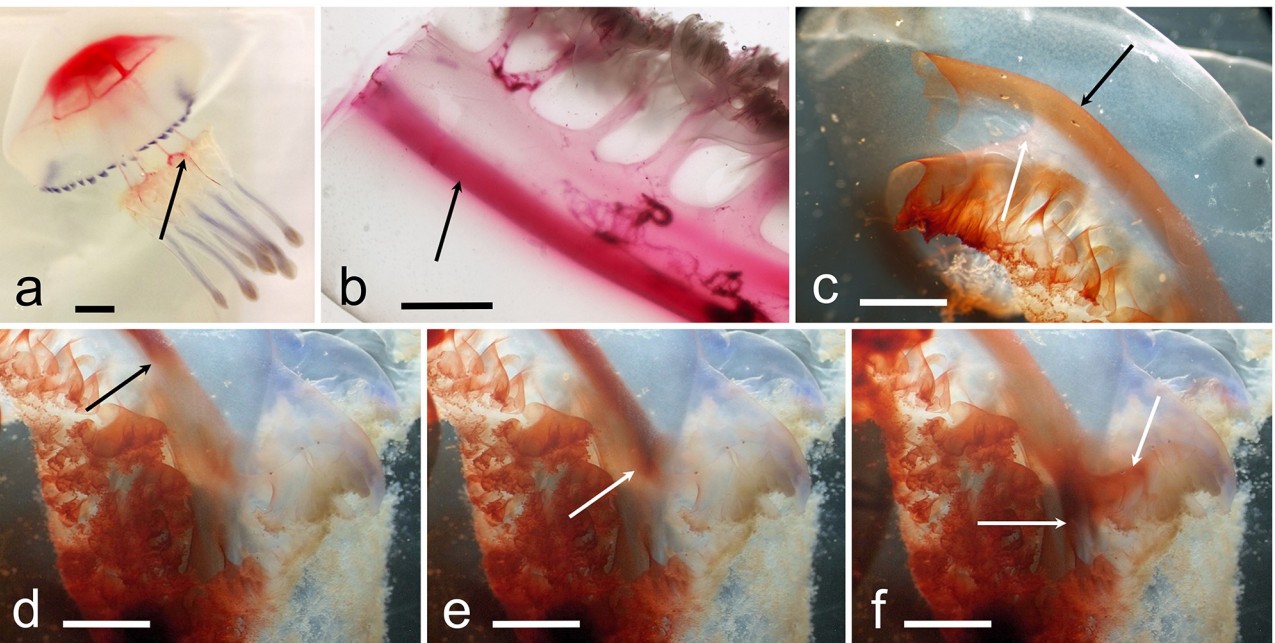

**Fig 10. Functional anatomy of the gastrovascular system.** (a) Living specimen as in Fig 9a and 9b, with mucous stained filament expelled from the upper openings of the external wings of the oral arms (scale bar = 1 cm). (b) Live excised scapula with red stain released just near the cutting surface (left). Stain mainly flowed in the lower hemi-canal (arrow) (scale bar = 0.5 cm). (c) Live oral arm cut under the scapulae, stained as above. Stain concentrated in the outer hemi-canal (black arrow), whilst the inner hemi-canal is unstained (white arrow). The frillings of the inner wing are stained by the stain droplets initially released in the proximity of the upper cutting surface, then externally diffused (bottom left) (scale bar = 0.5 cm). (d-f) Photographs from a clip following the stain diffusion into the outer hemi-canal. Arrows indicate three successive moments of the stain flux (scale bar = 0.5 cm).

upper portion of the external wings (Fig 10c–10f; S6 Video). An attempt to document the opposite flow by releasing the dye on the outer surface of the inner wing did not yield results, as the shock induced by the cut and related manipulation of the oral arm caused the release of mucus that quickly blocked the openings of all wings, internal wings included. Indeed, the experiment must be carried out in an extremely short time, in order not to lose an efficient flow induced by the ciliary beat, and therefore an "acclimatization" interval to reduce the phenomenon was not possible.

These data are supported by several *in vivo* observations, either in open water and in laboratory environment, where medusae, when physically stressed, quickly released a transparent semi-liquid stinging mucus from the outer wings of the oral arms and from the external portion of the scapulae. In contrast, in whole specimens, no mucus release from the inner wing of the oral arm has been recorded. It was also possible, during June-August, to observe some fully mature specimens releasing the eggs. In that situation, filaments of more compact mucus (even longer than twenty centimeters and sandy colored due to phytoplankton and suspended particles agglutination) were observed dangling from the outer wings only.

However, the presence of particles in the water allowed us to document their transit into the two hemi-canals, centrifugal in the external one, centripetal in the internal one (S7 Video).

The scapulae (living but excised) show a slower speed of the flux, anyway it was possible to detect an outgoing flux into the lower canal (Fig 10b), originating from the distal hemi-canal of the oral arm, which is in line with the previous observations in the oral arms canals.

The test carried out using living stained zooplankton as food for a 10 cm jellyfish gave results congruent with previous experiments, even if the dispersion of the plankton in the

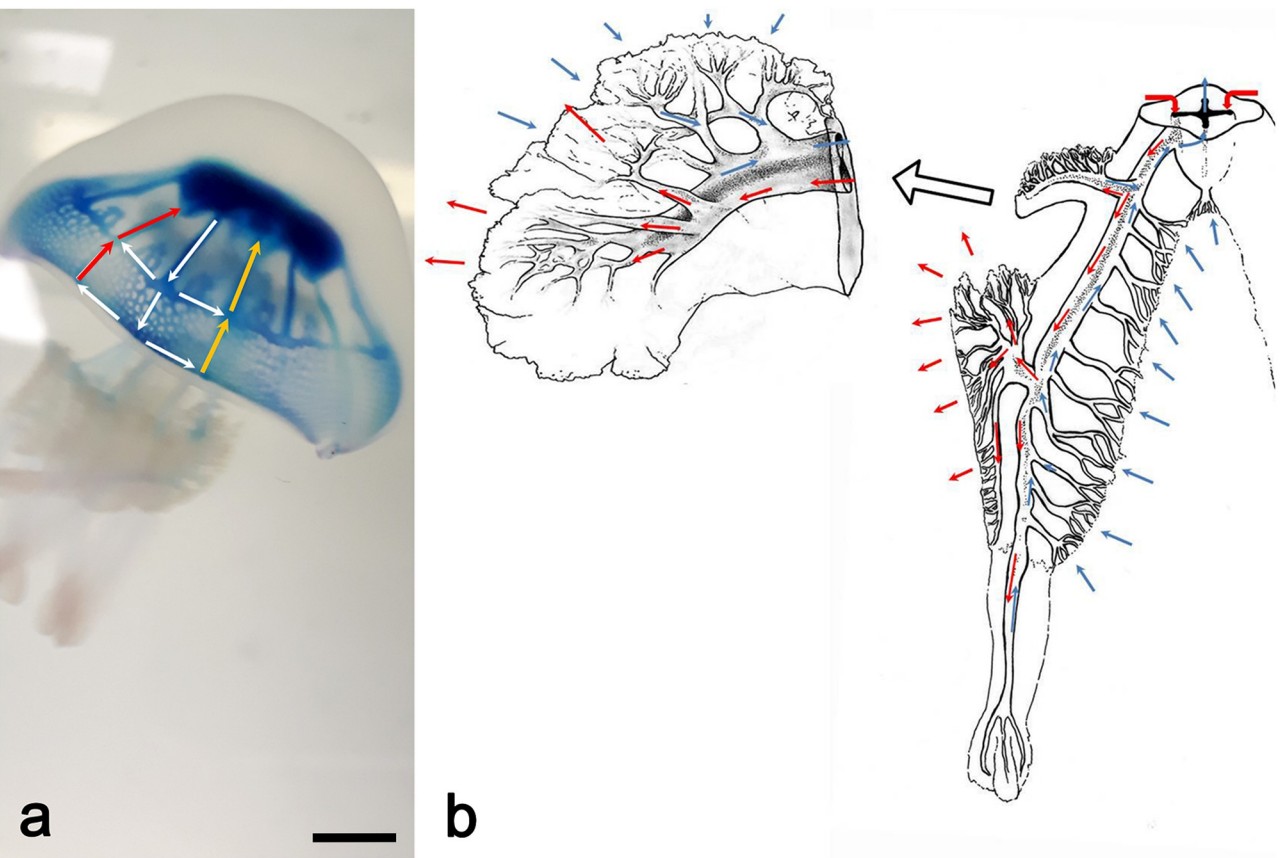

**Fig 11. Functional anatomy of the gastrovascular system.** (a) Living specimen as in Fig 9c–9e, with superimposed arrows that outline the flow sequence; I, from stomach to the inner ring canal through the adradial canals (white arrows); II, through the inner ring canal towards the per/interradial canals and towards the distal ring canal; III, through the distal ring canal towards the per/interradial canals (yellow arrows) and relative anastomoses; simultaneously a perradial outgoing flux flows from the inner ring canal junction towards the manubrium (red arrows) (scale bar = 1 cm). (b) Drawing showing the bidirectional flux in the manubrium. Blue arrows, incoming flux, red arrows, outgoing flux. Top left, magnification at scapular level.

body of water before its ingestion, and its fast transit in the inner hemi-canals (approximately 1/2–1 cm / sec), did not allow us to properly document this experiment, except as a personal observation.

As a whole, summarizing all our observations, it seems that the centripetal flow starts from the openings placed on the internal "conoid" of the oral arms (inner wings plus the openings of the central canal) and on part of the scapular medial upper openings connected to the superior hemi-canals in the scapulae, while the centrifugal flow affects the openings on the two external wings and the more distal upper scapular openings connected to the lower hemi-canals of the scapulae (Fig 11b).

## Discussion

Up to present day, the investigation of gastrovascular system in jellyfish mostly relied on the more traditional techniques: stain injection, dissections and external observations primarily on preserved specimens. Stain injections alone surely can highlight the path of canals trough the body of the jellyfish, but they can give few clues about the shape and the volume of the gastrovascular system. In addition, the traditional methods are significantly limited by the size and thickness of jellyfish. In large individuals, the remarkable opacity and thickness of the

mesoglea layer highly affect the quality and precision of the observations. That is why, currently, the shapes and representations of Rhizostomeae gastrovascular systems, based on this kind of observations, are not highly detailed and lack tridimensional data.

The protocol described in this work, based on resin endocasts + 3D imaging by X-ray microtomography, solves most of these issues. The resin endocast is relatively cheap and easy to reproduce, providing tridimensional information about the gastrovascular systems of jellyfish, not accessible with traditional methods. Once scanned, it can be further analyzed with dedicated software tools in order to extract quantitative parameters such as volume, length, thickness and connectivity of canals. The spatial resolution of microtomographic images allows the investigation of fine details or inaccessible areas, such as the count of branching and openings of *Rhizostoma pulmo* gastrovascular system, impossible to study with traditional methods. Furthermore, the virtual volume reconstructed via 3D X-ray imaging becomes easily sharable in multiple copies all over the world, thus avoiding the issues related to samples dimension, number, processing, and delivery. We intend to share on request the raw data obtained. Clearly, data have to be used only for scientific, educational or divulgative purposes (no lucrative use).

Additionally, our unexpected findings on the double flux circulation and the openings functional specialization shed new light on the knowledge about the appearance and development of trough-gut during evolution.

The genus *Rhizostoma* Cuvier 1800 includes three species, *R. pulmo* (Macri 1778), *R. octopus* (Gmelin 1791) and *R. luteum* (Quoy & Garmand, 1827). Their formal descriptions, however, lack of data about their gastrovascular system in the manubrium. Anatomic observations of this structure are only available for *R. octopus* [8, 30–32].

In, literature, the unique description of a similar hemi-canals system in the manubrium, as the one evidenced in our study, belongs to Stiasny (1921), which noted in *Lobonemoides robustus*, and other inscapulatae rhizostomeans, "two separate parallel tubes running inside the oral arm". Inside the genus *Rhizostoma*, he shortly reported in *R. octopus* "The wider canals usually show double lumen" [30]. Russell (1970) on the contrary, gave an accurate description of the canal system both in the umbrella and at manubrium level in *R. octopus*, as a unique canal. In his detailed studies, there is no mention of any medial narrowing in the manubrium canals. Considering the correctness of Russell (1970) description, the anatomies of the two species *R. octopus* and *R. pulmo* should differ [8].

*R. pulmo* showed this unique and never previously described two hemi-canal structure that runs along all the manubrium, excluding the terminal clubs. Apart the distal appendages, all the centrifugal circulation is completely physically separated from the centripetal one. In addition, no flow reversal was ever observed in our experiments.

The experiments on living specimens showed therefore a permanent specialized two-way circulation within the two hemi-canals: an outgoing flow in the distal hemi-canals expelled mainly from the upper external mouths of the upper portion of the oral arm, and an incoming flow starting from the frilled internal (medial) part of the oral arms, continuing in the proximal hemi-canals (Fig 11b) with an opposite direction compared with the parallel outgoing one.

Generally, in jellyfish, the oral mouth is thought to carry out both the ingestion and the egestion of food [6]. This is accomplished by spatially or temporarily separated currents that move food particles in and out the stomach to the mouth for excretion [13].

Classic umbrellar flows, observed in various jellyfish, mainly Semaeostomeae [8, 33–36], showed centrifugal flows exclusively into the adradial canals, with the interradial canals involved in a recirculation between the distal umbrella and stomach. Centripetal fluxes have been observed only in the perradial canals, flowing out at the four edges of the cruciform mouth in the Semaeostomeae (or the central cruciform canal in the Rhizostomeae). In *Aurelia*

*aurita* (Semaeostomeae), for example, the outward currents flow outside the gastrovascular system from the peripheral base of the gastro-oral groove while the inward currents flow in the proximal part of the groove [8, 13, 14].

Our observations of the umbrellar flow pattern in *R. pulmo* correspond broadly to that described in *A. aurita* [8] (apart from the differences related to the presence in *R. pulmo* of an internal ring canal and the numerous anastomoses present between the internal and external ring canals), with a centripetal flow exclusively in the perradial canals, physically isolated in their section between the internal ring canal and the central cruciform opening. Centrifugal flow occurs in the adradial canals, while the interradial canals appear to be involved in a recirculation from the periphery towards the stomach (Fig 11a).

In *Cassiopea* sp., a drop in the pH in association with the secretion of proteinase has been measured after feeding. When the digestion is completed, the gut is "flushed out" and the pH rises [37]. Based on literature, this mechanism could fit species like *Rhizostoma octopus* since a single channel through the manubrium is described [8]. However, it has to be considered that few studies investigated gastrovascular circulation in jellyfish, with the exception of studies focusing on feeding physiology [12, 17].

In *R. pulmo*, we showed a complete physical separation of the currents by the development of a layer between the opposite fluxes. This double canal structure also implied a differentiation of the function of the openings on the oral arms, that we will not call mouths anymore. The openings with the excretory function, "anuses", are only the ones placed on the external wings of the oral arms and part of the scapular ones. At the same time, the openings (now proper mouths) placed on the frilled internal (medial) wings of the oral arms have the inhaling function, considering that none of these internal openings was recorded as excretory. Furthermore, these observations match with the typical stinging mucus release behavior that barrel jellyfish perform into the wild when disturbed. Mucus is typically observed right on the outer surface of the oral arms, and on the most distal portion of the scapulae. The release of mucous strips containing oocytes also corresponded with the aforementioned observations.

It could be hypothesized that these directions of flow could be reversible but, in all our observations, a reversal of flows has never been recorded.

It is also interesting to focus on the results about the computation of number and thickness of anuses and mouths. The number of openings in the manubrium is apparently similar (1953 in vs 2048 out), plus, the canals leading to the mouths are slightly thicker if compared to those leading to the anuses. Considering a continuous double flux where the net outflow should be zero, the slightly greater thickness of mouths could balance in terms of circulation the higher number of smaller anuses.

In addition, this is the first quantification ever of the number of openings in scapulae and oral arms in Rhizostomeae. Up to now, in literature, they have been described just as "numerous", or "hundreds", or not reported at all [6–8, 13, 36].

It should be kept in mind that these values are not 100% accurate since the resin could have not filled all the canals (there are some indications that some of the canals of the outer wings, due to their limited thickness, are not always filled with resin up to all openings). This justifies the difference of the number of total openings based on the most complete single arm (4001) vs the number of total openings measured in the whole cast (2987, Table 1).

These observations are in contrast with the classic idea of Cnidaria anatomy and physiology where a single pore (or more openings, as in *R. octopus*) accomplish both uptake and excretion due to an intermittent (all in/all out) ciliary flux [10–12, 26, 37–39].

Anatomically speaking, the umbrellar canal system corresponds to Russell's observations in *R. octopus* [8], but it is not the case for the oral arm canals. There, the circulation pattern seems an adaptation towards a functional through-gut, since it has been shown to sustain a

permanent continuous flow and to support a continuous feeding since different openings have different functions. This new arrangement may advantage *Rhizostoma pulmo* in predation, growth and overall fitness since it may sustain a higher metabolism rate compared to other jellyfish characterized by intermittent feeding [40]. This could also be at the basis of the dimensions of *R. pulmo*, that is one of the biggest jelly species in the world in terms of biomass, which weigh even more than 25 kg [41].

From a broader perspective, our results may also give new clues about this adaptive development towards a sort of digestive apparatus analogous to the trough-gut common within bilaterians.

Generally, cnidarians are considered to possess a blind gut [6]. Our findings, together with Arai (1989) who showed radial canals distal excretory pores and papillae in *Aequorea victoria*, demonstrate that this paradigm is not 100% safe from debate [17]. In our case, we cannot define our results on the circulation in *Rhizostoma pulmo* as a proper through-gut but, surely, we can partially refuse the bias of a single oral opening with both the uptake and excrete function.

We don't believe that the whole classic paradigm is wrong, but that, in our case, this is rather a further example of adaptive convergence (by modifying pre-existing structures) in the direction of an anatomically different apparatus, but functionally analogous to a through-gut, just like the independent event within the *Aequorea* complex, resulting in analogue function-based structures in Cnidaria [42, 43]. Citing Dunn (2015) in his work "The hidden biology of sponges and ctenophores", we totally agree with his reflection that "early animal evolution has been presented as a ladder, where 'primitive' living species are thought of as the ancestors of 'complex' living species [44]. It is more and more evident that we cannot array animals from simple to complex, because there is no single axis of complexity".

As a conclusion, evolutionary scenarios cannot be easily solved but their comprehension has to be gathered step by step. In this context, deeper knowledge about the through-gut is fundamental to reconstruct the history of body plan evolution and diversification. Thus, further research is needed to discover, describe, examine and distinguish the various functional and structural similarities in Cnidaria and the other metazoan phyla. Within Rhizostomeae, further analyses will be needed to see whether similar adaptations are exclusive to *R. pulmo*, or are typical of the whole genus, and whether similar adaptations are present in other rhizostomeans, as Stiasny's (1921) pioneering observations would seem to indicate [30].

## Materials and methods

### Sampling area

*Rhizostoma pulmo* (Macri, 1778) samples (Table 2) were collected in the Gulf of Trieste (North Adriatic Sea, Italy), where two coastal sites were selected: Grado Lagoon and Duino-Sistiana (Gulf of Trieste) (Fig 12); in the south coastal area of France (Bages Sigean Lagoon and Thau Lagoon) and in the Gulf of Cadiz (specimens of *R. luteum*, atlantic coast of Spain). Dates of samplings are reported in Table 2.

### Sampling design

The choice of the two Italian areas derived from past observations of the research team. During summer (end of July/early August), blooms of young *R. pulmo* medusae were usually observed in the Grado Lagoon while blooms of adult medusae of *R. pulmo* were commonly observed through the Gulf of Trieste from summer to winter, as also reported by Andreja Ramšak and Katja Stopar (Pers. Observation, 2007).

**Table 2. List of collected samples and locations for *Rhizostoma pulmo* and *R. luteum*.**

| Species | Location | Geographical coordinates | Date of sampling | Collected by |
|---|---|---|---|---|
| *Rhizostoma pulmo* | Duino/Sistiana Gulf of Trieste (Italy) | 45˚45'42.2"N 13˚37'18.3"E | 08/03/2018 | Avian M., Motta G., Macaluso V., Pillepich N. |
| | | | 09/15/2019 | |
| | | | 07/09/2020 | |
| | | | 07/23/2020 | |
| | | | 08/26/2020 | |
| | | | 10/28/2020 | |
| | | | 07/28/2021 | |
| | Grado Lagoon (Italy) | 45˚40'59.4"N 13˚22'03.7"E | 07/31/2019 | Avian M., Motta G., Macaluso V., Pillepich N. |
| | | | 08/13/2019 | |
| | | | 10/11/2019 | |
| | | | 08/11/2020 | |
| | | | 08/16/2020 | |
| | | | 07/29/2021 | |
| | | | 09/02/2021 | |
| | Bay of Piran (Slovenia) | 45˚31'05.3"N 13˚33'09.5"E | 08/23/2018 | Ramšak A. |
| | | | 09/06/2018 | |
| | Thau Lagoon (France) | 43˚23'26.1"N 3˚36'12.0"E | 09/2018 | Bonnet D. |
| | Bages Sigean Lagoon (France) | 43˚05'51.6"N 2˚59'59.7"E | 06/27/2018 | Bonnet D. |
| *Rhizostoma luteum* | ICMAN Inst. P.Real, Cadiz (Spain) | - | 06/11/2019 | Prieto L. |
| | Bay of Cadiz (Spain) | 36˚32'38.8"N 6˚14'40.3"W | 07/09/2019 | Prieto L. |

Taking advantage from these observations, it was possible to collect specimens of all sizes, from juveniles to adults. All the samples were collected through diving sessions using containers of adequate size to avoid damages to medusae.

A total of 75 juveniles (diameter up to 15 cm) and 60 adult jellyfish (diameter greater than 15 cm) were locally sampled and brought to the lab. Here, half of the young specimens were reared in tanks (Aurelia-80, Reef-Eden), the remaining individuals were fixed in 4% formalin-saltwater solution and then stored in containers of adequate size. A total of six adult specimens came from France and a total of 30 lab-reared post-ephyrae while one young specimen of 4 cm diameter of *R. luteum* came from Spain (Table 1). These last specimens allowed us to start some preliminary comparative observations, which will lead to future research to compare *R. pulmo* with the other co-generic species.

## Morphological analyses

**Characterization of the gastrovascular system.** Different methods were tested to characterize the complex gastrovascular system of *Rhizostoma pulmo*.

**Contrast experimental methods.** First, different contrast media, such as Indian ink, neutral red 1.5%, carmine red 1.5%, 0.5% methylene blue and, in one case, 1% agar solution contrasted with 0.3% methylene blue have been injected to highlight the gastrovascular canals of both umbrella and manubrium. Injections have been made with an insulin syringe into the stomach from the exumbrellar apex. To maintain agar solution fluidity, samples, water and the containers were kept at a temperature of 60˚ C. For this analysis, formalin preserved individuals were employed. This type of experimental procedure has been used once only, as the endocast obtained is temporary (it lasts a maximum of a few days), and too fragile to be isolated from the surrounding tissues. All other experiments were performed with stained aqueous solutions.

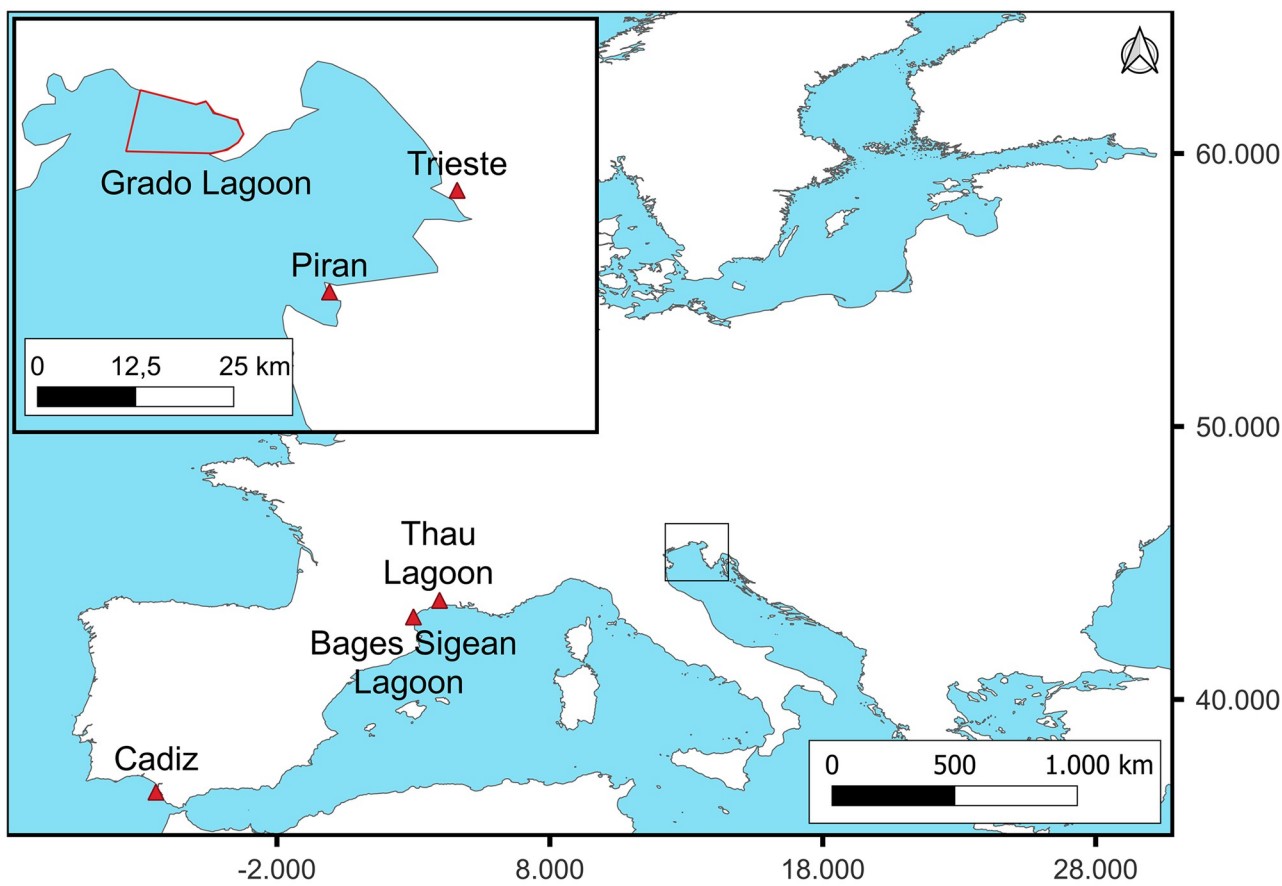

**Fig 12. Sampling map.** Square indicates the top left magnification of the Gulf of Trieste (Italy). Red triangles indicate the sampling areas. The red bordered area is part of the Grado Lagoon, where juvenile specimens were sampled.

Second, in order to evidence the gastrovascular circulation on living juvenile specimens of *Rhizostoma pulmo*, 2 ml of neutral red 1.5% in freshwater, or methylene blue 0.5%, were injected from the exumbrella apex with an insulin syringe into the stomach, letting the stain flow into the canals (12 experiments, performed between 2019 and 2020). Circulation was video recorded with a Canon G16 camera. Only living juveniles (collected at Grado Lagoon) were used due to their tissue transparency, adult tissues were too opaque to properly see through. After the staining phase, samples were quickly fixed in 4% formalin in seawater solution, then the scapulae and oral arms were excised and observed under a Leica Microsystems M205C Stereomicroscope equipped with a Canon G16 camera. The internal flux circulation of the seawater-stained solution in living specimens was video-recorded, and several pictures were taken.

Living zooplankton was also used to better understand the gastrovascular circulation. Living zooplankton was filtered through a 200 μm mesh zooplankton net, and stained with a vital seawater solution of methylene blue (0.3%) or neutral red (1%). Then, two medusae were separately fed with the stained zooplankton for a feeding time of 30 minutes; during this time interval samples were continuously observed. After feeding time, samples were fixed in formalin 4% solution and they were dissected for zooplankton identification in the canals of the gastrovascular system.

Furthermore, to better document the circulation process and the ciliary movement role, single oral arms from living samples were cut and vital-stained seawater solution was released near the mouth and canals openings at the section level. The same protocol was applied to investigate scapular circulation, where the stain solution was released in proximity of the cut surface of the scapulae. Inwards and outwards flows were simultaneously observed and recorded under a binocular microscope (Leica Microsystems M205C Stereomicroscope equipped with a Canon G16 camera).

**Internal cast of the gastrovascular system.** The aim of this experiment was to set-up a protocol to produce a cast of the whole gastrovascular system in order to understand the pattern of the complex canals ramifications. Several endocasts were made, either on whole specimens and on partially sectioned manubrium portions. For the following experiments formalin fixed individuals were used.

Three different types of resins were used:

a. Acrylic resin Mercox Blu (Ladd Research). Producer instructions were strictly followed. Resins were injected into the stomach (inner umbrella) of a 10 cm-diameter whole young specimen, and directly in an excised manubrium.

b. Epoxy resin (Resin 4 Décor©). Native resin was too dense to be easily injected without damaging the small channels. After several tests we found that diluting the resin with acetone at different concentrations to reduce viscosity (5% in bigger individuals and 7.5% in smaller ones) facilitated the injection without compromising the polymerization reaction.

c. Epoxy resin (Liquidissima, Resin Pro©). This resin was less viscous compared to the previous one, and a 5% dilution with acetone was enough to provide optimal results.

In whole organisms, injections were made in the stomach (through the exumbrella apex) with a 50 ml syringe and a rubber cannula. During the injection the organisms were kept in water and lightly massaged to allow the resin to flow all over the gastrovascular canals. After a hardening time of 24 hours, jellyfish were digested in a 20% solution KOH with a sample/solution volume ratio of 1:5 for 24 hours at 40˚C.

After several trials we can assert that the third resin was the most versatile tool, its reduced viscosity avoided excessive canal dilation and has ensured lower efforts during the injections. The second resin was the most viscous and required a greater dilution with the solvent to be efficiently injected. Anyway, both resins produced high quality endocasts. The first acrylic resin, due to its extremely low viscosity, specifically formulated for casts of circulatory systems, in our case it was not useful, as it failed to form a compact cast by fragmenting into a large number of shards.

**X-ray computed microtomography measurements.** The most complete endocast (of a young specimen of 14.6 cm of diameter, Grado Lagoon) was investigated by using the microfocus X-ray computed tomography (µCT) technique. The measurements were performed by using the FAITH instrument (manuscript in preparation) of the Elettra synchrotron light facility in Basovizza (Trieste, Italy). The FAITH station is a fully-customized cone-beam CT system equipped with a sealed microfocus X-ray source (Hamamatsu L12161, Japan) operating in an energy range from 40 to 150 kV at a maximum current of 500 µA. A 2192x1776 pixels flat panel detector (Hamamatsu C11701DK) featuring a pixel size of 120x120 µm$^2$ was used as detector. Exploiting the magnification effect offered by the cone-beam geometry [45], the source-object-detector distance can be varied to achieve a spatial resolution close to the minimum focal spot size of the source (5 µm) while imaging samples from a few millimeters up to about 20 centimeters in lateral size. The X-ray µCT scans were acquired using the following experimental conditions: tube voltage = 40 kV, tube current = 250 µA, number of

projections = 1800, angular step = 0.2 degrees, total scan duration = 6 min. The effective pixel size was set to 62.0 to image the whole endocast or selected regions of interest (local area modality), respectively. From the radiographs (projection images) acquired by the flat panel detector during a 360 degrees rotation of the specimen, a set of 2D axial slices was reconstructed by the free software NRecon 1.7.0.4 (Brulker, USA). In order to visualize and inspect the structure of the sample, the freeware software Fiji [46] and the commercial software VGStudio MAX 2.0 (Volume Graphics, Germany) were employed.

**Morphometric analysis of the X-ray microtomographic data.** To obtain a comprehensive description of the gastrovascular system, we analyzed two specific regions of the specimen, with a different focus: 1) the whole manubrium, to compare the structure of the oral arms vs. the scapulae. 2) A single oral arm, to characterize the topology of the three different structures (the wings) supporting the openings of the canals. For these analyses, the most whole and accurate casted and skeletonized oral arm has been chosen. This procedure ensured to avoid underestimation due to incomplete injection of the resin, cast fractures and, at the same time, to avoid under- or overestimation related with the skeletonization process.

The topological characterization of the systems features two main concepts: the local thickness (LT) analysis, and the "skeleton" (i.e., the medial axes of the structure, in this specific case) analysis. The LT of a segmented class in a tomographic dataset is described as the calculation for each voxel (vx) of the radius of the maximum inscribed sphere in the class that contains that vx [47]. The skeleton has been measured via the "thinning" algorithm [48], as we were interested in the calculation of the medial axis of the structures. The combination of the two analyses can provide a volume where the medial axes are labeled, vx by vx, with the LT values, to provide the local quantification of the thicknesses. A similar approach has been successfully applied to characterize the pore space topology in porous materials [49, 50].

The analysis on both the manubrium and the single arm was carried out as follows: first, the grayscale 8-bit dataset was segmented using the Otsu thresholding algorithm [51] applied to the full dataset grayscale frequency histogram to generate a binary volume. The binarized volume was then used to calculate the LT of the segmented class (the gastrovascular system). In the dataset #1 (manubrium) two sub-datasets, the first featuring the scapulae and the second featuring the 8 (eight) oral arms, were separated and analyzed independently. In the dataset #2 (the single oral arm) the three-winged portions were manually separated under the scapulae and subsequently independently analyzed.

The binary volumes of the different systems were filtered using an isotropic Gaussian filter with a structuring element of 5 (five) vx and followed by a segmentation preserving the original dataset connectivity to generate a smoother structure. This filtering is necessary to reduce the noise of the data and suppress the generation of many spurious branches in the skeletonization process. The 3D-isotropic-Gaussian filter fully preserves the mediality of the original dataset, while suppressing the smallest branches generated by the structure roughness during the thinning process. After the skeletonization, two consecutive 4 (four) vx pruning cycles, to delete the remaining short ending branches, were applied. This combination of filtering and pruning operations provided the smallest number of spurious end branches (e.g., the ones radiating from the long oral arm structures, generated by changes in shape and not by the presence of canals), while preserving the short branches at the oral apertures. In order to provide a thickness-labeled skeleton, each vx in the resulting skeletons was labeled using the respective LT value. The analysis of the single oral arm was carried a step further, to also investigate the number of openings in the three-winged portions: the number of endpoints (ideally equivalent to the number of oral apertures) was then calculated, separately for each wing of one oral arm.

**Morphometric analysis of jellyfish body (six from the Gulf of Trieste and six from the Bages Sigean and Thau Lagoon).** During this study several morphometric features of the

jellyfish were measured, but given the focus of the present work the results were here omitted. The diameter of a jellyfish is normally measured by taking the maximum diameter of the umbrella (ideally lying on a flat surface), between opposite rhopalia or between opposite marginal lappets. In the case of *Rhizostoma pulmo* (and many other rhizostomean jellyfish), it is impossible to stretch out its umbrella, so in order to take the "real umbrella diameter" (between two opposite rhopalia, or marginal lappets) using a soft measuring tape stretched over the exumbrellar surface is useful; but since in this species the umbrellar shape is practically a hemisphere, the functional *apparent diameter* (AD) was measured as AD = 2RD/π where RD is the *real diameter* (inter-rhopaliar semicircle, without the marginal lappets length). In the present study, all diameters reported refer to the AD, not the RD (S1a Fig).

## Supporting information

**S1 Fig. *Rhizostoma pulmo*, adult specimens, Gulf of Trieste, North Adriatic Sea.** (a) Semicircle indicates the real *umbrellar diameter*, straight line indicates the *apparent diameter*. (b) Upside-down specimen showing the three-winged pattern of the oral arms. White arrows indicate the two external wings, black arrow the internal one (Photo courtesy of Paolo Coretti). Both specimens have an *apparent diameter* of about 30 cm.
(TIF)

**S2 Fig. Early development of the manubrium canal system.** (a) Young specimen of 2.7 cm diam., subumbrellar view. Arrow indicates the residual of the central mouth opening still open (scale bar = 0.25 cm). (b) As in (a), specimen 3 cm diam. (scale bar = 0.25 cm). (c) Subumbrellar view of a 4 cm diam. specimen, manubrium excised under the genital sinuses. Visible the four perradial canals projecting into the still wide central canal (scale bar = 0.25 cm). (d) Same view in a specimen of 5 cm diam., with the same pattern as in (c), but with a noticeable size reduction of the central quadrangular canal (scale bar = 0.25 cm). Black arrows in (c) and (d) indicate the edges of the perradial canals, white arrows indicate the central quadrangular canal.
(TIF)

**S3 Fig. Central area of the manubrium and hemi-canals.** (a) Specimen of 12.4 cm diam. with the juvenile central mouth closed (arrow) (scale bar = 1 cm). (b) oral arm of a specimen of 23.5 cm diam., showing the stained hemi-canal system on the right (arrows) and on the left the anastomosis that gives rise to the canals that reach both the inner wing (bottom left) and the outer ones (top).
(TIF)

**S1 Video. Micro CT-scan 3D rendering.** 360° animation of the whole endocast evidencing the different structures: umbrella (green), scapulae (blue), central canal (yellow), oral arms (red).
(MP4)

**S2 Video. Micro CT-scan 3D rendering.** 360° animation of the whole endocast evidencing transverse sections of the manubrium.
(MP4)

**S3 Video. Micro CT-scan 3D rendering.** 360° animation of the whole endocast evidencing oblique, longitudinal, and transverse sections of the manubrium at supra-scapular and sub-scapular (at the distal portion of the central canal) level.
(MP4)

**S4 Video. Micro CT-scan 3D rendering.** 360˚ animation of the whole endocast evidencing transverse sections of the umbrella, manubrium at supra-scapular and sub-scapular (at the distal portion of the central canal) level and finally longitudinal sections at central canal level.
(MP4)

**S5 Video. Methylene blue stain diffusion into the gastrovascular system.** Specimen of 4.2 cm in diameter. The injection of the stain was carried out into the stomach, just in the middle of the exumbrella.
(MP4)

**S6 Video. Neutral red stain diffusion into the gastrovascular system of an excised oral arm.** The stain was released on the cutting plan (top). Internal wing on the left, the external ones on the right.
(MP4)

**S7 Video. Bidirectional particle flux into the hemi-canals.** The oral arm is the same sample of Video 6.
(MP4)

## Acknowledgments

The authors would like to thank Dario Radin, Trieste, Italy, which actively supported with his boat several samplings in the Gulf of Trieste and in the Grado lagoon, Italy. We acknowledge Elettra Sincrotrone Trieste for providing access to its laboratories and we thank the staff for assistance in using the FAITH instrument.

## Author Contributions

**Conceptualization:** Massimo Avian.

**Data curation:** Massimo Avian, Lucia Mancini, Marco Voltolini, Diego Dreossi, Vanessa Macaluso, Nicole Pillepich, Laura Prieto, Andreja Ramšak, Gregorio Motta.

**Formal analysis:** Massimo Avian, Lucia Mancini, Marco Voltolini, Gregorio Motta.

**Investigation:** Massimo Avian, Delphine Bonnet, Diego Dreossi, Vanessa Macaluso, Nicole Pillepich, Laura Prieto, Andreja Ramšak, Antonio Terlizzi, Gregorio Motta.

**Methodology:** Massimo Avian, Lucia Mancini, Marco Voltolini, Vanessa Macaluso, Nicole Pillepich, Gregorio Motta.

**Resources:** Delphine Bonnet, Gregorio Motta.

**Software:** Lucia Mancini, Marco Voltolini.

**Supervision:** Massimo Avian.

**Writing – original draft:** Massimo Avian, Lucia Mancini, Marco Voltolini, Delphine Bonnet, Vanessa Macaluso, Nicole Pillepich, Laura Prieto, Andreja Ramšak, Antonio Terlizzi, Gregorio Motta.

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
