## [Decision Letter · Decision Letter 0]

18 Apr 2022

PONE-D-22-08636A novel endocast technique providing a 3D quantitative analysis of the gastrovascular system in Rhizostoma pulmo: an unexpected through-gut in CnidariaPLOS ONE

Dear Dr. Avian,

Thank you for submitting your manuscript to PLOS ONE. After careful consideration, we feel that it has merit but does not fully meet PLOS ONE’s publication criteria as it currently stands. Therefore, we invite you to submit a revised version of the manuscript that addresses the points raised during the review process.

Both reviewers found your study imformative and technically sound, but they have raised a number of important points that you will need to address satisfactorily. Particular attention should be paid to what findings of your study are novel and which ones were already known.  You also need to avoid overstating the impact of your findings, as pointed out by Reviewer 2.  ==============================

We look forward to receiving your revised manuscript.

Kind regards,

Robert E. Steele, Ph.D.

Academic Editor

PLOS ONE

Journal Requirements:

2. In your Methods section, please provide additional information regarding the field permits you obtained for the work. Please ensure you have included the full name of the authority that approved the field site access and, if no permits were required, a brief statement explaining why.

4. We note that Figure 12 in your submission contain [map/satellite] images which may be copyrighted. All PLOS content is published under the Creative Commons Attribution License (CC BY 4.0), which means that the manuscript, images, and Supporting Information files will be freely available online, and any third party is permitted to access, download, copy, distribute, and use these materials in any way, even commercially, with proper attribution. For these reasons, we cannot publish previously copyrighted maps or satellite images created using proprietary data, such as Google software (Google Maps, Street View, and Earth). For more information, see our copyright guidelines: http://journals.plos.org/plosone/s/licenses-and-copyright.

a. You may seek permission from the original copyright holder of Figure(s) [#] to publish the content specifically under the CC BY 4.0 license.  

5. We note you have included a table to which you do not refer in the text of your manuscript. Please ensure that you refer to Table 2 in your text; if accepted, production will need this reference to link the reader to the Table.

Reviewers' comments:

Reviewer's Responses to Questions

**Comments to the Author**

1. Is the manuscript technically sound, and do the data support the conclusions?

Reviewer #1: Yes

Reviewer #2: Partly

2. Has the statistical analysis been performed appropriately and rigorously? 

Reviewer #1: Yes

Reviewer #2: Yes

3. Have the authors made all data underlying the findings in their manuscript fully available?

Reviewer #1: Yes

Reviewer #2: Yes

4. Is the manuscript presented in an intelligible fashion and written in standard English?

Reviewer #1: Yes

Reviewer #2: Yes

5. Review Comments to the Author

Reviewer #1: This paper is novel and the findings provide important implications to the study of micro-anatomy and differential development of function and organ system in cnidarians. The method in this paper is obviously important for the related research in the future, I, therefore suggest to publish. Below are some concerns to be considered maybe in the revised manuscript or reply to reviewers.

I really like the novel method to endocast the jellyfish. I think this method could be applied to study other transparent or translucent animals not just jellyfish, e.g. ctenophores. However, I have some comments on the "a through-gut" of Rhizostomeae. As the authors stated: "The inflow involves only the “mouth” openings on the internal wing of the oral arm and relative inner hemi-canals while the outward flow involves only the two outermost wings’external hemi-canals and relative“anal”openings on the external oral arm. "; and indeed the innovative method clearly shows the double-direction of water flows. However, the author should firstly discuss this topic within the scope of cnidarians and then extend to the bilaterians, and then we will have a better understanding of the origin of such double-direction flows in jellyfishes. The double-direction water flows (in and out) driven by gastric cillia in siphonoglyph, are well known in gastric cavity of anthozoans. The hemi-canal flows, equivalent to the roof and floor flows, are well known in other jellyfishes. The radial canals seen in living jellyfishes, however, were absent in Cambrian benthic medusozoans (Han et al., 2016; Wang et al., 2017), and were most probably derived from the reduction of broad gastric pouches by closing of endoumbrella and exumbrella(endodermic perradial fusion in Han et al., 2016). The arm canals of Rhizostomeae, were developed from the closure of the oral groove. Thus the "through gut with oral anus", is not strange in cnidarians and should be a specialized structure independently acquired in Rhizostomeae.

Han, J., Kubota, S., Li, G.X., Ou, Q., Wang, X., Yao, X.Y., Shu, D.G., Li, Y., Uesugi, K., Hoshino, M., Sasaki, O., Kano, H., Sato, T., Komiya, T., 2016. Divergent evolution of medusozoan symmetric patterns: Evidence from the microanatomy of Cambrian tetramerous cubozoans from South China. Gondwana Research 31, 150-163.

Wang, X., Han, J., Vannier, J., Ou, Q., Yang, X., Uesugi, K., Sasaki, O., Komiya, T., Sevastopulo, G., 2017. Anatomy and affinities of a new 535-million-year-old medusozoan from the Kuanchuanpu Formation, South China. Palaeontology 60, 853-867.

Reviewer #2: In this paper, Avian et al. use resin endocasts to describe the gastrovascular system of the jellyfish Rhizostoma pulmo. The beautiful endocast imagery is paired with detailed morphometric work. It was interesting to see what kind of data could be collected from the endocast, but most of the details are not directly pertinent to the main argument in this manuscript—that there is directional flow in the hemi-canals and therefore a through-gut. Despite the emphasis on novel endocast techniques and 3D X-ray computed microtomography, the main argument is primarily supported by more traditional stain injection experiments (Figures 9-11 and the associated videos). Unfortunately, I was unable to take that data and reconstruct the model the authors provide. I detail my concerns below; if they can address these issues, I would find the paper markedly stronger:

1) I am personally unable to see how the authors go from the photo in Figure 9a to the model in 9b. A photo of comparable parts of the oral arm might help. I watched the associated video but was still unable to tell what data to take from it.

2) Figure 10c is one of the more compelling images. It shows how stain is concentrated in the outer hemi-canal but not the inner hemi-canal. It is unclear though how this image relates to the staining experiments illustrated in Figs. 9-11. Is this part of the same set of experiments? Is it from one of the experiments where the oral arms were cut off? If so, it seems plausible that directional flow could be a behavioral response to tissue damage.

3) Assuming the model proposed here is correct, there is nothing in the manuscript to demonstrate that directional flow in the hemi-canals is related to food digestion. The movement of dye demonstrates how water circulates through the animal, but how important this current is to the intake or excretion of food remains unknown. It therefore feels premature to call these pores “mouths” or “anuses” as the authors do.

4) My next point is a matter of stylistic choice. The authors use strong language to describe the significance of their results, often referring to the blind-gut model of cnidarian anatomy as a “paradigm” or “dogma” they have overthrown. They call their results “astonishing”, with “huge consequences” for our understanding of bilaterian gut evolution. But the authors never explain exactly how this new information from Rhizostoma impacts current hypotheses of bilaterian bodyplan evolution. As the authors note, anal pores have been described in ctenophores as well as the medusozoan Aequorea, so through-guts are not unheard of outside of the Bilateria. Presnell (2016), which is cited several times in this paper, serves as a bit of a cautionary tale. Although Presnell’s description of a through-gut in ctenophores is described in this manuscript as paradigm shaking, anal pores in ctenophores had already been reported by Louis Agassiz in 1850, and some scientists were surprised anyone found Presnell’s finding remarkable (see Tamm, Sidney L. "No surprise that comb jellies poop." Science 352.6290 (2016): 1182-1182). The authors are free to discuss their results in terms they choose, but I think most readers will find these results interesting but far from paradigm shifting.

5) The authors rightly mention that one benefit of their new technique is that “3D X-ray imaging becomes easily sharable in multiple copies all over the world”. In that spirit, I think this paper would be stronger if the authors provided the actual 3D model as a supplemental or otherwise accessible file.

6) One final, minor point. The authors call the Medusozoa “a remarkable homogeneous group.” I cannot think of anything further from the truth. Consider the dramatic differences in body plan between box jellies, siphonophores, and myxozoans, just to name a few!

6. PLOS authors have the option to publish the peer review history of their article (what does this mean?). If published, this will include your full peer review and any attached files.

Reviewer #1: **Yes: **Jian Han

Reviewer #2: No

---

## [Author Response · Author response to Decision Letter 0]

1 Jun 2022

Reply to the Editor

Dear E. Steele, we intend to follow your suggestion to deposit our lab. protocols in protocols.io, obviously if our work will be accepted.

The authors received no specific funding for this work.

Regarding point 4), we changed the source of satellite image contained in Fig. 12 from Google Earth to Natural Earth (Free vector and raster map data @ naturalearthdata.com). Thus, the map picture is now ok to be published. 

5) We have referred Tab 2 into the text, as you asked.

All figures were checked with PACE, now all meet PLOS requirements.

Just a question. To make the separate flow regime better understandable for reviewer # 2, apart from the explanations in this "Response to Reviewers", we have also tweaked something in our Results as well as Materials and Methods, and, in this regard, we have inserted a sentence (p 14 in Results, and p 22 in Materials and Methods) on one of the experiments carried out, which was omitted in the previous version, as it was not possible to document it with videos. It is therefore a question of personal visual observations. If you think it is best to omit this piece, we would leave the decision to you.

Reply to Reviewer #1

Dear Jian Han, thank you for your useful comments, which we have taken into account. Now, based on your suggestions, as well as those of reviewer #2, we've extensively reviewed the through-gut discussions, both in the Abstract, Introduction and Discussion. 

Reply to Reviewer #2

Dear reviewer #2, thank you for your comments and, in some cases, doubts. Regarding your doubts in interpreting our statements, points 1 - 3, below you will find our clarifications, and we have also revised some points in the Results, as well as in the Materials and Methods, to make our experiments and their meaning clearer, also adding a little something that was not in the previous version of the manuscript.

1-3) Fig 9a shows one of our first experiments with the injection of a dye solution (neutral red) into the stomach, approximately 60 sec after the injection, in exumbrellar view. We have put this picture to show how the dye diffuses in the eight adradial canals quickly reaching the inner ring canal. Fig. 9b shows the same specimen, but in lateral view, 1 min after figure 1, to highlight how the interradial canals begin to bring the dye back towards the stomach. This photo also shows how part of the dye injected into the stomach is also spreading into the outer hemi-canals of the oral arms, reaching the upper part of the external wings. Fig 9c is comparable to 9a, in a further experiment, with a different dye solution (methylene blue, has a much more intense staining capacity, and therefore the injected solution is much more diluted, and is better diffused by ciliary currents. The solution of neutral red, or that of carmine red, were much more concentrated, "oily", and diffused more slowly and with difficulty. For this reason, apart from the first couple of experiments, we used methylene blue, even if the red ones were more contrasted). The following photos are all of the specimen in 9c, in three successive stages, as in the video. These are just some of the experiments we carried out (we did multiple stain diffusion tests in summer 2019, and a further set in 2020), and as a whole they allowed us to reconstruct the flows which were then schematized in Fig.11a and 11b . Fig. 10a is the same specimen of Fig. 9a and b, used here to indicate one of the colored mucus filaments now “pushed out” from the upper openings of the external wings.

The dissection of the specimens used in the experiments, after fixation, allowed us to understand which of the hemi-canals of the oral arms were involved in the centrifugal flow, and they were always the external ones, as indicated in the scheme of Fig. 11b. But the intensity of the residual colors after fixation (after all the dyes used are vital, and therefore labile) were too weak to obtain good photos, so we proceeded to set up another set of experiments, on living parts, but dissected, of larger specimens, observable at higher magnification under a stereomicroscope. These tests have been documented in the images of Fig. 10 b-f, and related videos. These experiments have shown that: the ciliary flows, even after dissection, are maintained for a few minutes and with the same modality and directionality observed in the specimens as a whole, and as can be seen in video 6 and 7. In the movie 7, single particles present in the water, penetrate into the hemi-canals of the oral arm, both from the cut area and also from the inner wing openings (left in the video), pass in opposite directions, and those directed towards the stomach are present in the internal hemi-canal. All these data are congruent with in-sea observations, where the jellyfish, when stressed by external pressure waves caused by the movement of someone nearby, release mucus (transparent but still visible), with unpleasantly stinging characteristics, mainly from the upper and external part of the outer wings and from the distal part of the scapulae, not from the inner wings. Further observations (not included in the initial manuscript, now added) made in the summer months, during egg release events, highlighted strips of mucus, more compact, even 20 cm long and more (which soon agglutinate the phytoplankton and suspended particles present in the seawater, taking on a sandy coloration) which remain dangling from the outer wings in a relatively tenacious way, even for a long time. Also in this case these strips are located exclusively on the outer wings.

We also set up an experiment to document the flow of food collected from the oral arms in whole specimens, using living zooplankton that was filtered through a 200 μm mesh zooplankton net, and stained with a vital seawater solution of neutral red (1%). Then, medusae were fed with the stained zooplankton for a feeding time of 30 minutes, after waiting about 15 min after transferring the jellyfish to a beaker to disperse the released mucus. Unfortunately, the low density of the colored organisms, their small size, the low contrast intensity of the dye, and an unsuspected speed of transit of the single prey in the visible portion (between the end of the scapulae and the beginning of the wings) of the oral arm canals (approximately 1/2 - 1 cm/sec) did not allow us to document in an appreciable way the transit of preys inside the canals of the oral arms, difficult to focus in such a short time in a mobile jellyfish. For these reasons, this experiment was initially not reported in the manuscript. The naked eye observations, however, coincide with the data obtained in previous experiments. Now we are inserted concisely these observations.

From all these results, we had a solid basis to describe the flow, and therefore also the related transport of the preys, in the system of double canals. In the hemi-canals, at the level of the oral arms, only the internal hemi-canal showed a centripetal flow. The distinction between "mouths" and "anuses" is based on the direction of the flow, which on the basis of our data is fixed, continuous and not reversible, not even in sectioned parts.

It’s true that digestion does not take place in the manubrium, but, as already known, inside the stomach and its adjacent umbrellar canals. However, since this species feeds from filtering seawater from its openings, as well as all rhizostomeans, it is evident that internal circulation is strictly related with food ingestion and distribution.

4) Regarding the observation of point four, we have followed these comments, as well as the ones made by reviewer 1, and these parts are now substantially modified into the ms.

5) The reviewer is right, and we have no problem sharing the data obtained, on the contrary, we would be happy. Clearly, data have to be used only for scientific, educational or divulgative reasons (no lucrative use). So, we will share them freely after a simple collaboration request, with the clause of citing the sources. Now this availability has been inserted in the ms.

6) We agree with the reviewer that the medusozoans are an apparently incredibly heterogeneous group, from the point of view of observable body plans. But from the phylogenetic point of view they are however considered monophyletic, meaning as Medusozoa the clade Staurozoa + Cubozoa + Scyphozoa and the Hydrozoa. The so-called Myxozoa would be part of the Endocnidozoa (Myxozoa + Polypodiozoa), which, at least at the moment, are considered a sister group of the Medusozoa.

One of the several papers on this argument is:

Kayal E., Bentlage B., Pankey M.S., Ohdera A.H., Medina M., Plachetzki D.C., Collins A.G., Ryan J.F., 2018. Phylogenomics provides a robust topology of the major cnidarian lineages and insights on the origins of key organismal traits. BMC Evolutionary Biology 18:68

With the hope of having clarified your doubts, anyway we remain at your disposal for any further request for clarification, if necessary.

on behalf of all co-authors,

yours sincerely,

Massimo Avian

---

## [Decision Letter · Decision Letter 1]

12 Jul 2022

A novel endocast technique providing a 3D quantitative analysis of the gastrovascular system in Rhizostoma pulmo: an unexpected through-gut in Cnidaria

PONE-D-22-08636R1

Dear Dr. Avian,

We’re pleased to inform you that your manuscript has been judged scientifically suitable for publication and will be formally accepted for publication once it meets all outstanding technical requirements.

Kind regards,

Robert E. Steele, Ph.D.

Academic Editor

PLOS ONE

Additional Editor Comments (optional):

I am happy to make the decision to accept your manuscript. Note that Reviewer 2 suggests two minor edits to the text. I believe these suggestions clarify the work and ask that you make these edits when you submit the final version of your manuscript. I agree with Reviewer 2 regarding putting your data in a public database, but I will not make this a requirement for acceptance.

Reviewers' comments:

Reviewer's Responses to Questions

**Comments to the Author**

1. If the authors have adequately addressed your comments raised in a previous round of review and you feel that this manuscript is now acceptable for publication, you may indicate that here to bypass the “Comments to the Author” section, enter your conflict of interest statement in the “Confidential to Editor” section, and submit your "Accept" recommendation.

Reviewer #1: All comments have been addressed

Reviewer #2: All comments have been addressed

2. Is the manuscript technically sound, and do the data support the conclusions?

Reviewer #1: Yes

Reviewer #2: Yes

3. Has the statistical analysis been performed appropriately and rigorously? 

Reviewer #1: Yes

Reviewer #2: Yes

4. Have the authors made all data underlying the findings in their manuscript fully available?

Reviewer #1: Yes

Reviewer #2: No

5. Is the manuscript presented in an intelligible fashion and written in standard English?

Reviewer #1: Yes

Reviewer #2: Yes

6. Review Comments to the Author

Reviewer #1: I am satisfied with the authours's substantial revision according to my suggestion, I am glad to see this article published in PLoS One.

Reviewer #2: The changes the authors have made address the concerns that I had. Congratulations on an interesting read!

I would still prefer that the authors make the raw data available on a public database instead of providing it upon request. I will leave it up to the editor to decide what is appropriate in this case.

In addition, I have two minor suggestions that the authors are welcome to accept or not:

1) In the abstract and the discussion, the authors try to nuance their main argument by invoking their specific taxon (i.e. "the gastrovascular system in Rhizostoma...partially invalidates the paradigm (at least for this taxon) of a single oral opening with both the uptake and excrete function"; "...we can partially refuse the bias (at least in this taxon) of a single oral opening with both the uptake and excrete function"). I appreciate the attempt at nuance, but I think this phrasing doesn't quite work, since it is already known that Rhizostomae lack a single oral opening. I think removing the parenthetical will make these sentences clearer.

2) In my initial review, I disagreed with the authors' description of Medusozoa as a "homogenous" group, given the high degree of morphological diversity. The authors countered that the Medusozoa are monophyletic, and wrote a sentence in the paper to state as much ("Regarding gut anatomy and digestive mechanisms, the Subphylum Medusozoa, even if comprising morphologically heterogeneous taxa, from the phylogenetic point of view it is anyway considered monophyletic"). I personally think this sentence reads awkwardly, and would be confusing to a reader who wasn't familiar with our back-and-forth. I certainly never meant to suggest that medusozoans are not monophyletic. Perhaps the following change?:

"The Medusozoa are a monophyletic groups with a wide range of gut anatomies and digestive mechanisms."

7. PLOS authors have the option to publish the peer review history of their article (what does this mean?). If published, this will include your full peer review and any attached files.

Reviewer #1: **Yes: **Jian Han

Reviewer #2: **Yes: **David A. Gold

---

## [Editor Report · Acceptance letter]

26 Jul 2022

PONE-D-22-08636R1 

A novel endocast technique providing a 3D quantitative analysis of the gastrovascular system in *Rhizostoma pulmo*: an unexpected through-gut in cnidaria 

Dear Dr. Avian:

I'm pleased to inform you that your manuscript has been deemed suitable for publication in PLOS ONE. Congratulations! Your manuscript is now with our production department. 

Kind regards, 

on behalf of

Dr. Robert E. Steele 

Academic Editor

PLOS ONE